# Impact of maximal overexpression of a non-toxic protein on yeast cell physiology

Yuri Fujita[1], Shotaro Namba[1], Yoshiaki Kamada[2,3], Hisao Moriya[4]*

[1]Graduate School of Environmental, Life, Natural Science and Technology, Okayama University, Okayama, Japan; [2]National Institute for Basic Biology, Okazaki, Japan; [3]Graduate University for Advanced Studies (SOKENDAI), Hayama, Kanagawa, Japan; [4]Faculty of Environmental, Life, Natural Science and Technology, Okayama University, Okayama, Japan

## eLife assessment

This **convincing** study advances our understanding of the physiological consequences of the strong overexpression of non-toxic proteins in baker's yeast. The findings suggest that a massive protein burden results in nitrogen starvation and a shift in metabolism likely regulated via the TORC1 pathway, as well as defects in ribosome biogenesis in the nucleolus. The study presents findings and tools that are **important** for the cell biology and protein homeostasis fields.

*For correspondence:
hisaom@okayama-u.ac.jp

Competing interest: The authors declare that no competing interests exist.

## Abstract

While it is recognized that excess expression of non-essential proteins burdens cell growth, the physiological state of cells under such stress is largely unknown. This is because it is challenging to distinguish between adverse effects arising from the properties of the expressed excess protein (cytotoxicity) and those caused solely by protein overexpression. In this study, we attempted to identify the model protein with the lowest cytotoxicity in yeast cells by introducing a new neutrality index. We found that a non-fluorescent fluorescent protein (mox-YG) and an inactive glycolytic enzyme (Gpm1–CCmut) showed the lowest cytotoxicity. These proteins can be expressed at levels exceeding 40% of total protein while maintaining yeast growth. The transcriptome of cells expressing mox-YG to the limit indicated that the cells were in a nitrogen source requirement state. Proteome analysis revealed increased mitochondrial protein and decreased ribosome abundance, similar to the inactivated state of the TORC1 pathway. The decrease in ribosome abundance was presumably due to defective nucleolus formation, partially rescued by a mutation in the nuclear exosome. These findings suggest that massive overexpression of excess protein, termed protein burden, causes nitrogen source starvation, a metabolic shift toward more energy-efficient respiration, and a ribosomal biosynthesis defect due to an imbalance between ribosomal protein and rRNA synthesis in the nucleolus.

## Introduction

Expression levels of proteins are optimized to maximize the fitness of organisms, and the evolutionary principle that determines those levels is considered to be demand and constraint (**Bruggeman et al., 2020**; **Figure 1A**). Demand is the evolutionary pressure that works to increase the expression of a protein to meet the requirements of its function, and constraint is the evolutionary pressure that works to decrease the expression of a protein to avoid the negative effects produced by excess protein. Constraints appear as disturbances in cellular function when the protein is overexpressed (**Figure 1A**, the red arrow). The mechanisms of constraining expression levels, or defects due to overexpression, can be roughly classified into four major categories (**Moriya, 2015**; **Figure 1B**).

'Resource overload' (also called 'burden'; *Kastberg et al., 2022*) is a disorder in which intracellular resources used for protein processing, such as synthesis, localization, degradation, and modification of proteins, are depleted or monopolized by the processing of that protein, resulting in a disruption of processing of other proteins (*Dong et al., 1995*; *Eguchi et al., 2018*; *Kintaka et al., 2020*; *Kintaka et al., 2016*; *Stoebel et al., 2008*). This phenomenon occurs in proteins with high expression levels. 'Stoichiometry imbalance' is a disorder triggered by an imbalance of amounts between constituents within complexes, leading to untimely activation or inactivation (pathway modulation), degradation (resource overload), aggregation (promiscuous interaction), etc. (*Kaizu et al., 2010*; *Makanae et al., 2013*; *Papp et al., 2003*). This occurs due to the overexpression of proteins that form complexes. 'Pathway modulation' is a disorder in which pathways are untimely activated or inactivated. This occurs especially with overexpression of proteins involved in pathway regulation, such as metabolic enzymes, kinases and phosphatases, and transcription factors (*Prelich, 2012*; *Youn et al., 2017*). 'Promiscuous interaction' is a disorder in which weak interactions are enhanced by mass action to form complexes or aggregates that are not normally formed. This damages cellular function by inactivating essential proteins or overloading degradation resources and is caused by overexpression of intrinsically disordered proteins or liquid-liquid phase-separating proteins (*Bolognesi et al., 2016*; *Ma et al., 2010*; *Vavouri et al., 2009*).

As the expression level of a protein increases, it eventually encounters one of the barriers created by either these specified constraints or others that are unknown, all of which are related to the function and physicochemical properties of the protein and damage cellular function. The expression level at which this occurs is referred to as the 'expression limit' in this paper (see *Figure 1A*). If the constraints could be removed from the protein, the expression limit would increase. For example, if components of a complex are simultaneously expressed to resolve a stoichiometry imbalance, the expression limits of those proteins will be increased (*Kaizu et al., 2010*; *Makanae et al., 2013*). In this manner, when (one) constraint is removed from a protein, the expression limit would increase until the next constraint barrier is hit (*Figure 1A*, dotted lines). The amount of expression limit created by a constraint barrier depends on the type of constraint. Stoichiometry imbalances, pathway modulations, and promiscuous interactions would create different expression limits for each protein. This is because they are determined by the activity, regulatory mechanism, and physicochemical properties of each protein. On the other hand, resource overload is presumed to be determined to some extent by the capacities of the processes in the cell (*Kintaka et al., 2020*; *Figure 1C*). Essentially, the synthesis, in which all proteins are processed, should have the largest capacity, while processes such as transport, folding, and degradation, in which only a fraction of proteins are processed, should have smaller capacities. In fact, adding transport or degradation signals to a cytosolic protein lowers the expression limit (*Kintaka et al., 2016*). In that light, the final constraint barrier that emerges when all possible constraints are removed is the overloading of the synthesis process.

Overloading of the synthesis process is specifically referred to as 'protein burden' or 'protein cost' (*Eguchi et al., 2018*; *Kafri et al., 2016*). It has long been the subject of study, mainly as an effect on growth produced by excess (gratuitous) proteins that are not necessary for cellular function (*Bruggeman et al., 2020*; *Scott et al., 2010*). This idea was first proposed in studies of the cost of unnecessary induction of the *lac* operon in *Escherichia coli* (*Koch, 1983*). In recent years, much attention has been paid to the actual mechanisms by which massive expression of excess protein inhibits growth and the response of cells (*Farkas et al., 2018*; *Kafri et al., 2016*; *Metzl-Raz et al., 2020*). During protein burden, it is presumed that depletion of the materials, machinery, and factors necessary for transcription and translation occurs, along with competition from these monopolies. In particular, amino acids and aminoacyl-tRNAs are major candidates for depletion (*Kastberg et al., 2022*). In addition, ribosomes, which are crucial for protein synthesis, themselves require significant resources for their synthesis and have therefore been considered major candidates for depletion during protein burden, as they compete for these limited resources (*Metzl-Raz et al., 2017*). Mechanistic analysis of protein burden in budding yeast, a model eukaryotic cell, has been intensively conducted by Barkai and colleagues, who reported that transcription is rate-limiting in phosphate starvation and translation in nitrogen source starvation (*Kafri et al., 2016*), and that basic transcription factors are limiting resources (*Metzl-Raz et al., 2020*). In addition, a cellular response in the form of increased cell size and enhanced protein synthesis capacity has been observed (*Kafri et al., 2016*). Thus, the previously expected decrease in ribosomes has not been detected.

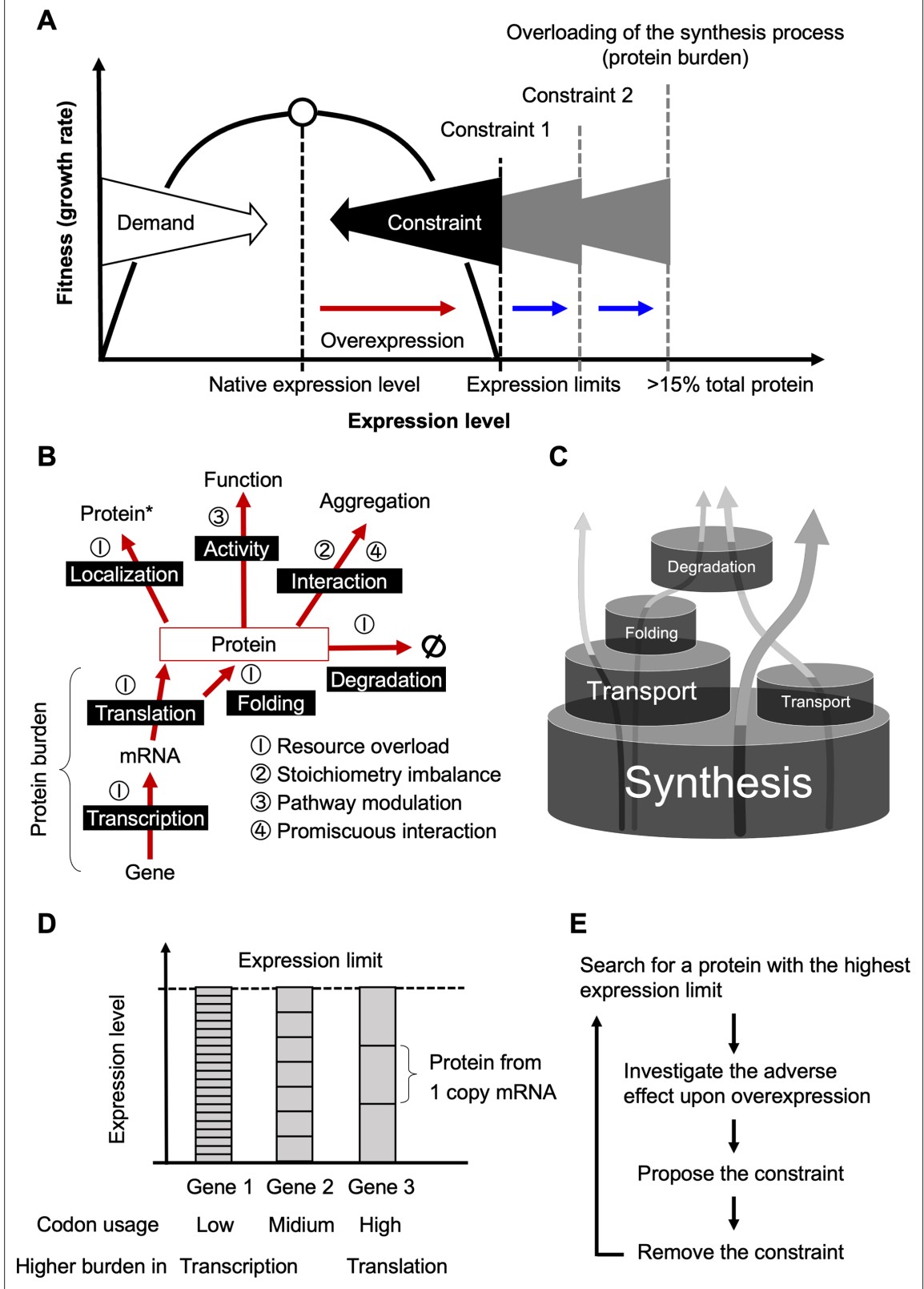

**Figure 1.** Schematic diagrams illustrating the constraints on protein expression levels. (**A**) Protein expression levels were determined by demand and constraints (created by the authors, inspired by the work by ***Keren et al., 2016***). The existence of this evolutionary principle is revealed by the relationship between fitness (growth rate) and expression level when the expression level is altered, and in general, the native expression level provides the highest level of fitness for the organism (***Bruggeman et al., 2020***). The existence of constraints is revealed by a decrease in fitness upon

*Figure 1 continued on next page*

*Figure 1 continued*

overexpression (red arrow). Constraints determine the expression limit of the protein, that is the expression level at which growth inhibition occurs (dotted lines), but there can be multiple constraints for a single protein. The constraint that exists at the highest expression limit is protein burden. (**B**) Four major mechanisms that constrain protein expression levels. The figure outlines the fate of proteins and shows what adverse effects occur upon overexpression (red arrows). The resource overload that occurs within the synthesis process (transcription and translation) is specifically referred to as protein burden. This diagram is a more concise redrawing of the author's earlier one (*Moriya, 2015*). That asterisk means differentially localized proteins. (**C**) A barrel model for resource overload (modified from the author's earlier one) (*Kintaka et al., 2020*). The size of the barrels represents the capacity of each process, and the arrows represent the fate of the proteins processed there. The expression limit of a protein is defined by the lowest-capacity process that processes the protein. Among these processes, synthesis, which processes all proteins, is considered to have the largest capacity. Thus, proteins processed only by synthesis would have the highest expression limits, and their overexpression would overload synthesis (i.e. cause protein burden). Previous studies by the authors, conducted in this context, estimated that protein burden occurs at more than 15% of total protein (*Eguchi et al., 2018*). (**D**) Difference in burden due to gene structure (codon optimization). The amount of transcription required to achieve the same protein expression limit depends on the degree of codon optimization. This may change whether transcription or translation results in a higher burden. (**E**) An ideal framework for protein burden studies. See text for details.

As mentioned above, to investigate protein burden, we need to overexpress proteins with minimal constraints. This is because overexpressing proteins with additional constraints cannot overload the synthetic process, as the additional constraints reduce the expression limits (*Figure 1A*). Also, the cellular damage and cellular response generated by the constraint mask the protein burden response. However, it remains unclear which proteins are not subject to constraints other than protein burden. Proteins that would induce protein burden have been determined by assuming that they would be minimally constrained based on their properties, and this assumption has been confirmed experimentally. Model proteins selected in this manner in yeast include fluorescent proteins (FPs) and certain glycolytic enzymes (*Eguchi et al., 2018*; *Farkas et al., 2018*; *Kafri et al., 2016*; *Kintaka et al., 2016*; *Kintaka et al., 2020*). FPs do not have a specific function or interaction partner in yeast cells and are not actively sorted into cellular compartments other than the cytosol. On the other hand, glycolytic enzymes are naturally expressed at very high levels and are thus considered to be less constrained. Notably, these proteins exhibit an expression limit of approximately 15% of total protein in yeast cells (*Eguchi et al., 2018*). However, it is not clear whether these proteins are free from constraints other than protein burden. This is because there may be unknown constraints, and there is no evidence that the measured expression limit is the 'true limit' at which protein burden occurs.

In addition to the protein (amino acid sequence) itself, there is another aspect to be considered when selecting a model for protein burden. That is the nucleotide sequence of the gene used for expression, that is transcriptional, and translational regulatory sequences, codon usage, and so on (*Figure 1D*). For example, if a protein is to be expressed to the limit by a less codon-optimized and thus less translationally efficient mRNA, a large amount of mRNA would be required, which would induce a greater burden on transcription. Conversely, a similar attempt by an mRNA with high translation efficiency would induce a greater burden on translation. These are the constraints inherent in protein synthesis itself. Although there is some evidence that protein synthesis in nature is economically designed to optimally load transcription and translation (*Frumkin et al., 2017*; *Hausser et al., 2019*; *Mahima and Sharma, 2023*), it still needs to be determined what the genetic design should be to minimize the burden.

Considering the above circumstance, a realistic approach to protein burden requires more than just drawing straightforward conclusions from the overexpression of a single model protein (or gene). Rather, it requires experimentally identifying proteins with the highest possible expression limits, investigating the negative impacts of their overexpression, and, if any constraints are revealed, examining whether their removal can increase expression limits. Through successive iterations, this process gradually deepens our understanding of protein burden (*Figure 1E*). In each iteration, the least constrained protein (and its gene) will be generated and utilized as a benchmark for further investigation into proteins with high (or low) expression limits and the adverse effects of their overexpression. This will (unintentionally) elucidate a new mechanism of constraint. Indeed, the enhanced green fluorescent protein (EGFP), which we have used as a model for protein burden (*Kintaka et al., 2020*; *Kintaka et al., 2016*; *Makanae et al., 2013*), was found to form a chaperone-entrapped aggregate upon overexpression and trigger the heat shock response due to its low folding ability (*Namba et al., 2022*). Its overexpression also causes abnormal yeast cell morphology through proteasomal stress as it contains cysteines. The moxGFP (mox), which improved folding properties and lacks cysteine

residues (*Costantini et al., 2015*), showed a higher expression limit and no longer induced abnormal morphologies. However, even when overexpressing mox, the anticipated depletion and competition for factors, or cellular responses to them have not been observed (*Namba et al., 2022*).

In this study, we found that a mutation causing loss of fluorescence in mox and other fluorescent proteins increased the expression limit up to threefold, to more than 40% of total protein. Introducing a neutrality index (NI) to assess protein constraint, we found that non-fluorescent mox (mox-YG) and enzymatically inactive Gpm1 (Gpm1–CCmut) are currently the least constrained. Upon overexpression of mox-YG, depletion of amino acid or nitrogen sources, decreased ribosomal expression, and a metabolic shift from glycolysis to oxygen respiration were observed. Thus, physiological states, as anticipated with such protein burdens, are only generated by massive overexpression of proteins with significantly lower constraints. Some of the responses found appear to be due to inactivation of the TORC1 pathway, suggesting that eukaryotic cells use this regulatory pathway to buffer protein burden. In the burdened cells, the nucleolus was dysplastic, and mutations in the nuclear exosome partially restored this. Thus, the abnormal formation of liquid-liquid phase-separated organelles due to dilution may be one of the defects caused by the strong protein burden.

## Results

### Fluorescent property is a constraint on the expression of fluorescent proteins

To evaluate the expression limits of FPs, we overexpressed FPs to levels that induced growth inhibition. Gene constructs, in which the *TDH3* promoter controls the target protein genes, were incorporated into plasmids designed for the genetic-tug-of-war (gTOW) method (*Figure 2A*, *Figure 2—figure supplement 1A*; *Moriya et al., 2006*). Subsequently, yeast cells were transformed and cultured in a medium depleted of leucine. The maximum growth rate (MGR) of cells was calculated from serial measurements of optical density at 660 nm ($OD_{660}$). Protein levels were determined through the analysis of proteins from cells in the logarithmic growth phase (*Figure 2—figure supplement 1B*). For each protein, we also analyzed a mutant in which the Tyr residue forming the fluorophore was replaced with Gly (YG mutation), resulting in a loss of fluorescence. This is because we unexpectedly found that this mutation substantially increased the expression limits of FPs (see below).

Overexpression of the FPs and mutants all caused growth inhibition (*Figure 2B and C*). EGFP and mox were expressed at about 15% of the total protein as previously reported (*Kintaka et al., 2016*), but mCherry (mChe) and YG mutants were expressed at higher levels. Mox-YG expressed 44% of the total protein (*Figure 2D and E*). Our previous work had evaluated only expression limits (*Eguchi et al., 2018*), but we realized that we could not properly assess the constraints on each protein if the growth rates at the expression limit were different. For example, the mChe's expression limit (20%) is higher than that of sfGFP (17%), but the relative MGRs are 20% and 80%, respectively, which is much lower for mChe. The relationship between %MGR and %protein level when the control is set at 100% is shown in the scatter plot in *Figure 2F*. The lower left indicates more constraint as lower expression levels lead to growth inhibition, and the upper right indicates less constraint as higher expression levels do not cause growth inhibition. Here, we describe 'unconstrained' as 'neutral' and calculate the neutrality index (NI) as the product of %MGR and %protein level.

Note that the regime of NI in *Figure 2F* shows a theoretical neutrality distribution but does not show the relationship between growth rate and the expression level when the expression level of a protein with a particular neutrality is increased or decreased. For example, 20% of the growth rate would be maintained when the expression of mox-YG (NI = 2400) is increased to 100%, but this is not realistic. We also note that in this study, we calculated NI using the percentage of the target protein among total cellular proteins separated by SDS-PAGE. Therefore, to calculate NI, the expression limit of the target protein must be sufficiently high to be detectable by this method. On the other hand, if detection sensitivity is improved, for example, by using western blotting, it becomes possible to estimate the NI even for proteins under strong constraints—that is, those that exhibit cytotoxicity. In fact, when we recalculated the NI for EGFPs fused with various localization signals from our previous study (*Kintaka et al., 2016*), all values were found to be below 100 (*Figure 2—figure supplement 3*).

The NI of each protein indicates that introducing a non-fluorescent mutation in FPs uniformly increases neutrality (*Figure 2G*), suggesting that 'having fluorescence' is a constraint on the expression

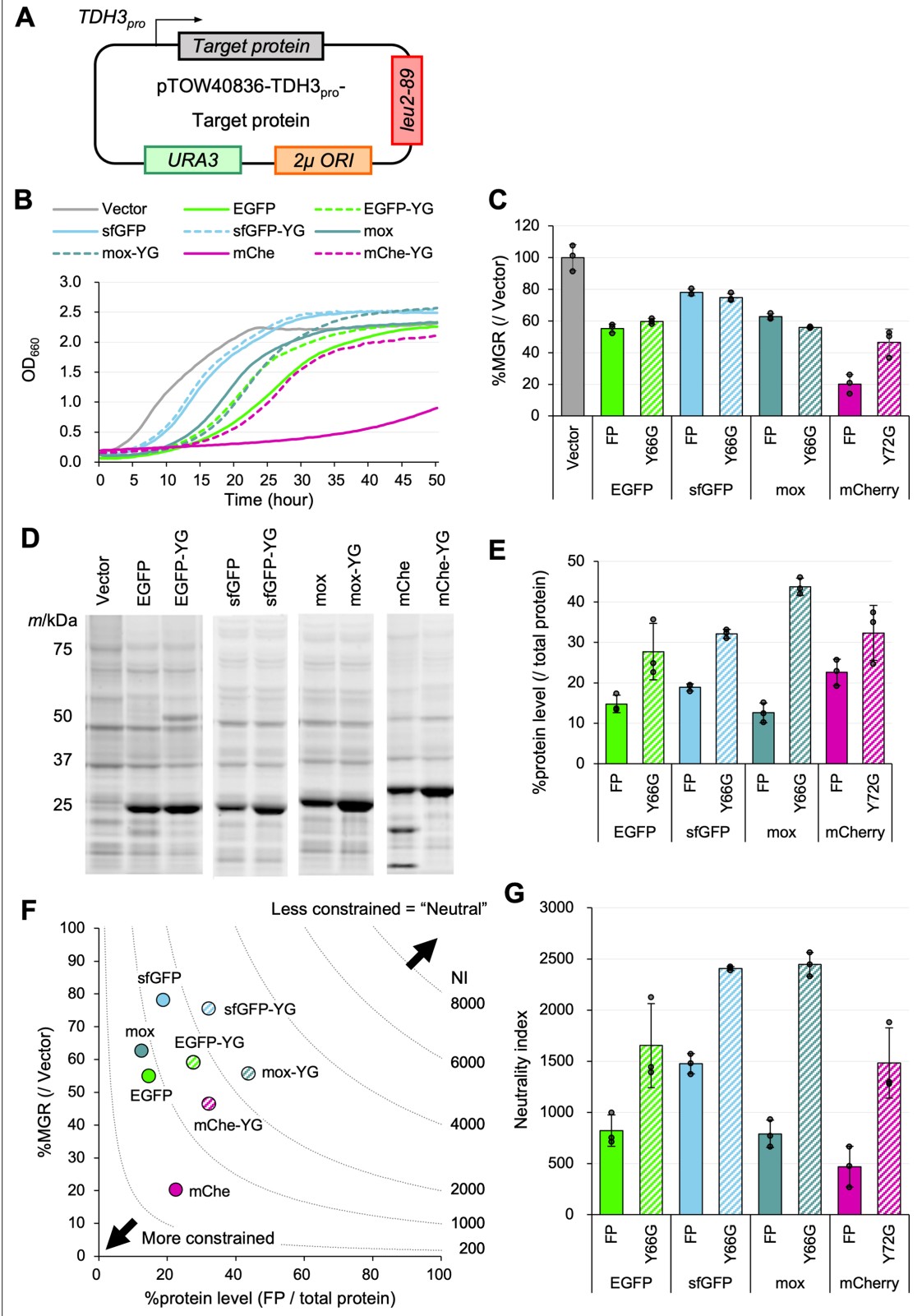

**Figure 2.** Evaluation of expression constraint (or neutrality) of fluorescent proteins. (**A**) Experimental setup of the analysis. Target proteins were expressed under the control of the *TDH3* promoter (*TDH3pro*) using the multicopy plasmid pTOW40836. The cells were pre-cultured under –Ura conditions, and then cultured in –LeuUra conditions. (**B**) Growth curves of the vector control and cells overexpressing FPs and their mutants in synthetic medium (–LeuUra). The solid or dotted lines show the average calculated from three biological replicates. Growth curves with the standard deviations

*Figure 2 continued on next page*

*Figure 2 continued*

(SD) of replicates are shown in *Figure 2—figure supplement 2*. (**C**) MGR of cells overexpressing FPs and their mutants (percent over the vector control). (**D**) Gel images of SDS-PAGE-separated proteins extracted from cells overexpressing FPs. All proteins were separated after staining with a fluorescent dye. (**E**) Protein level of the target protein (percent over the total protein). The amount was calculated from the intensity of the target bands separated by SDS-PAGE of D. See *Figure 2—figure supplement 1B* for the method of FP and total protein quantification. Note that EGFP-YG and mCherry may not be accurately measured due to dimerization and cleavage of the protein, respectively. (**F**) Relationship between MGR and the protein level. MGR data are the same as in C, and the protein level data are the same as in E. The neutrality indexes (NIs), calculated from the products of MGR and protein levels, are indicated by the lines. If the investigated proteins have the same NI, they are located on the same dotted line segment. (**G**) The neutrality index for the FPs and mutants. The neutrality index is the product of %MGR (/Vector) and %protein level (/total protein). The bars and error bars in C, E, and G show the means and SDs calculated from three biological replicates. The raw data is shown with dot plots.

The online version of this article includes the following source data and figure supplement(s) for figure 2:

**Source data 1.** Original files for SDS-PAGE analysis displayed in *Figure 2D*, *Figure 2—figure supplement 5C*, *Figure 2—figure supplement 7G*, *Figure 2—figure supplement 8B*, and *Figure 2—figure supplement 10D and H*.

**Figure supplement 1.** Methods used for overexpression and protein expression quantification.

**Figure supplement 2.** Comparison of growth of fluorescent protein overexpressing strains.

**Figure supplement 3.** Neutrality indexes of constrained proteins.

**Figure supplement 4.** Base and amino acid sequences of moxGFP and sfGFP.

**Figure supplement 5.** Comparative evaluation of mCherry (mCherry-Kafri) used in previous studies and this study.

**Figure supplement 6.** Overexpression of mox-YG does not cause cell enlargement or burden on transcription.

**Figure supplement 7.** Phototoxicity is not the constraint of mox expression.

**Figure supplement 8.** Effect of Tyr and oxidative stress on the expression limits of mox and mox-YG.

**Figure supplement 9.** Overexpressed mox-YG does not show any specific localization.

**Figure supplement 10.** Mutational analysis of mox.

of FPs. The most neutral proteins were sfGFP-YG and mox-YG, with NIs of about 2400. Notably, despite significant differences in the degree of growth inhibition and expression limits between the two proteins, their NIs are nearly identical (*Figure 2F and G*). They differ only in two amino acid residues (Cys - Ser; *Figure 2—figure supplement 4*), which seem to generate differences in expression efficiency. However, this does not appear to affect their NIs.

We also investigated the impact of expression efficiency on NI with another FP, mChe. The mChe used in this study is codon optimized (the codon adaptation index (CAI)=0.584; *Keppler-Ross et al., 2008*), while the mChe used by Kafri et al. for protein cost analysis (here referred to as mChe-Kafri) is not codon optimized (CAI = 0.236; *Kafri et al., 2016*). When overexpressed, mChe-Kafri (and mChe-Kafri-YG) had lower expression and growth inhibition than mChe (and mChe-YG; *Figure 2—figure supplement 5A–D*). This may be due to the lower translation rate of mChe-Kafri mRNA. However, their NIs were not significantly different, especially for the YG mutation (*Figure 2—figure supplement 5E, F*). Thus, the constraints indicated by NI are apparently not affected by translation efficiency. In the study using this mCherry-Kafri, it was also reported that the more mChe-Kafri was expressed, the more the cells enlarged (*Kafri et al., 2016*). However, no cell enlargement was observed with mox-YG, which can be expressed more than mCherry. Interestingly, the mChe-YG cells did not show enlargement (*Figure 2—figure supplement 6A, B*), indicating that cell enlargement is a phenomenon that occurs specifically when fluorescent mChe is overexpressed.

It should be noted that the growth inhibition caused by the overexpression of mox-YG in this study is a burden on translation rather than on transcription. This is because overexpression of a frameshift mutant (moxFS), in which four nucleotides were inserted directly after the ATG of the mox-YG gene, abolished the notable growth delay (*Figure 2—figure supplement 6C, D*).

## Constraint on fluorescent protein expression arises from fluorophore properties

Next, we further analyzed the possibility that fluorescence constrains the expression of FPs. When yeast was grown in the dark, the degree of growth inhibition of mox was the same (*Figure 2—figure supplement 7A*) and the expression limit of mox was rather increased (*Figure 2—figure supplement 7B*). Phototoxicity does indeed exist when expressing mox with fluorescence. This is because, when

exposed to strong light, cells overexpressing mox show a reduced growth rate (*Figure 2—figure supplement 7C–E*) and abnormalities in their internal structures (*Figure 2—figure supplement 7F*). However, the expression limits of mox and mox-YG were almost unchanged by the light (*Figure 2— figure supplement 7G, H*). Thus, the constraint was not due to phototoxicity itself.

The number of Tyr residues was also irrelevant because NI was like that of mox-YG when a single Tyr was added to the C-terminus of mox-YG (*Figure 2—figure supplement 8A–E*). Reactive oxygen species ($H_2O_2$) are generated during fluorophore formation. When $H_2O_2$ was added to the medium, stronger growth inhibition occurred only in mox (*Figure 2—figure supplement 8F–H*). This suggests that the reactive oxygen species generated during fluorophore formation are responsible for the constraint. However, this possibility is not supported by the omics analysis described later (see Discussion). Mox-YG, like mox, was present throughout the cytoplasm and did not cause any characteristic localization or aggregation (*Figure 2—figure supplement 9*).

During this study, we examined a moxGFP (mox-T203I) in which Thr203 was accidentally replaced by Ile (*Figure 2—figure supplement 10*). The fluorescence of this mutant is reduced to 78% of mox (*Figure 2—figure supplement 10A–C*). The growth rate of the cells expressing this mox-T203I was like that of mox and mox-YG (*Figure 2—figure supplement 10E*). Still, the expression limit was intermediate between mox and mox-YG (*Figure 2—figure supplement 10F*), resulting in the NI intermediate between mox and mox-YG (*Figure 2—figure supplement 10G*). This result further supports that fluorescence is related to constraints. However, as mentioned above, fluorescence emission itself should not be toxic (*Figure 2—figure supplement 7*). We next focused on the formation process of the fluorophore: the tyrosine of the GFP fluorophore undergoes phenol and phenolate state changes, with Thr203 stabilizing the phenol form and substitution to Ile reducing the phenolate form (*Kummer et al., 2000*). Furthermore, in enhanced GFPs such as mox, substitution of Ser65 before Tyr66 for Thr anchors the phenolate form. To investigate the possibility that this is a factor determining the limit of mox, we produced mox with Thr65 replaced with Ser and examined its expression limit. The mox-T65S mutant was expressed 12% more than mox (*Figure 2—figure supplement 10H, I*), suggesting that the phenol-type fluorophore can be a limiting factor for the expression level of the fluorescent protein, while the concrete mechanism is still unclear.

## Overexpression of mox-YG causes an amino acid starvation response

To understand the cell physiology under protein burden, we next performed transcriptome (RNA-seq) analysis of the cells overexpressing mox-YG, the least constrained protein. Although the expression limit of mox-YG is about threefold higher than that of mox (*Figure 2E*), the growth rates at the expression limits of both are not largely different (*Figure 2C*). Thus, by comparing transcriptomes upon their overexpression, we may be able to distinguish and extract the general transcriptional response due to reduced growth rates from the (protein burden) response associated with the massive expression of a less constrained protein.

The expression changes under mox and mox-YG overexpression were calculated as fold change (FC) over the vector control (*Figure 3—figure supplement 1A, B*). The overall trend of expression changes was similar ($r=0.64$, *Figure 3A*). This similar variation may include responses associated with the expression system and reduced growth rate. On the other hand, there were also different transcriptional responses between the two. Gene categories that showed significant expression changes in either mox or mox-YG overexpression are shown in *Figure 3B*. In addition to the enhanced ribosome synthesis and decreased glycolysis observed in previous mox overexpression (*Namba et al., 2022*), enhanced oxidative phosphorylation was detected. Interestingly, 'Ribosome biogenesis', which was elevated in mox overexpression, showed a rather decreasing trend in mox-YG overexpression (*Figure 3B*). Indeed, *IFH1*, which is involved in ribosomal gene transcription (*Martin et al., 2004*), was upregulated in mox, but downregulated in mox-YG (*Figure 3A*). Different trends were also observed for 'amino acid metabolism' (*Figure 3B*). The amino acid transporter genes *AGP1* and *GAP1* were upregulated only in mox-YG overexpression (*Figure 3A and C*). Expressions of specific amino acid and nitrogen source transporter genes were strongly induced in mox-YG (*Figure 3C*). The expression of glucose transporter genes was also very different (*Figure 3C*). These results suggest that mox and mox-YG induce similar but distinct transcriptional responses, particularly that mox-YG overexpression may result in depletion of amino acid and nitrogen sources and decreased ribosome synthesis.

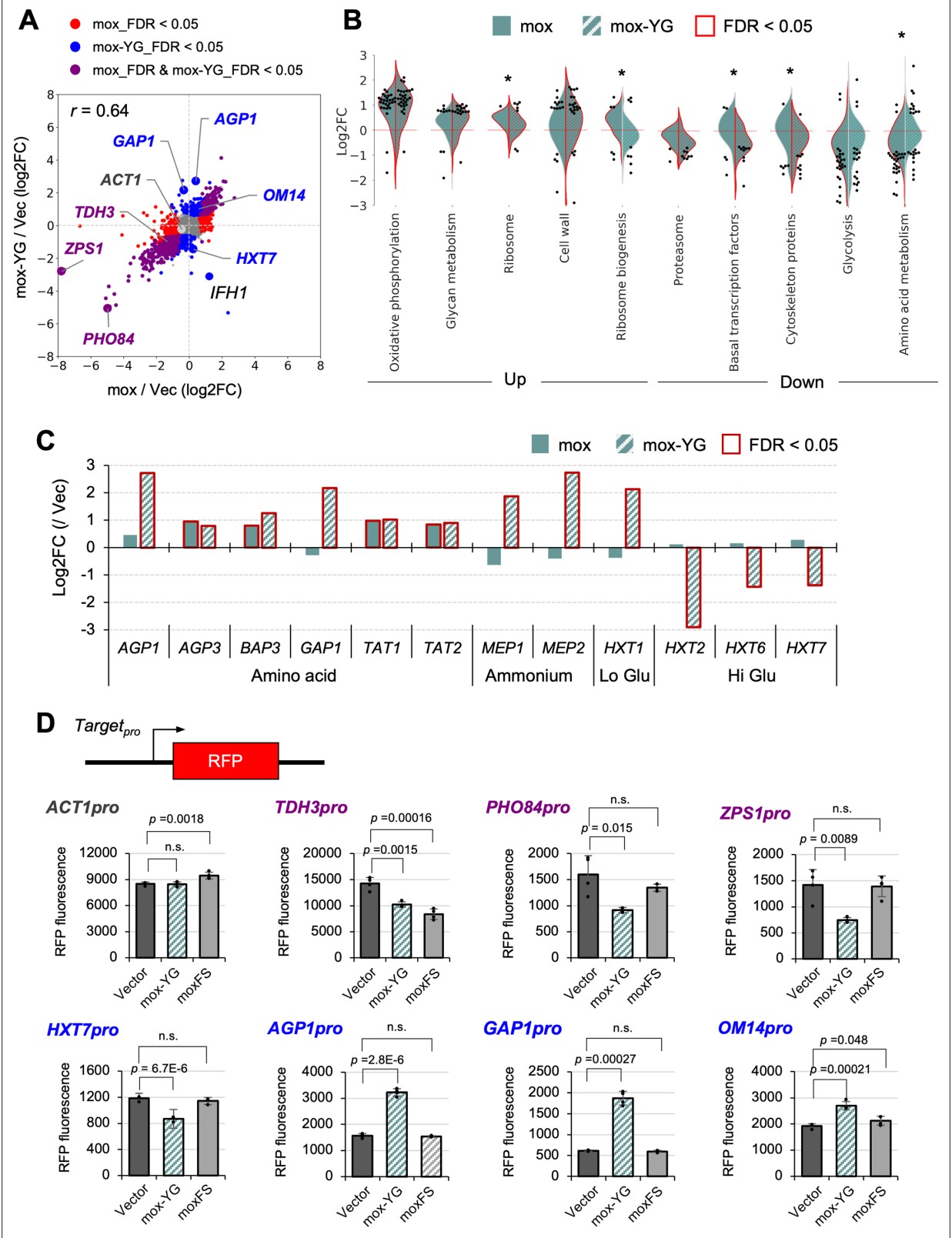

**Figure 3.** Transcriptional response of mox and mox-YG overexpression. (**A**) Comparison of transcriptional responses of mox and mox-YG overexpression. Genes that showed common or different transcriptional responses in mox and mox-YG overexpression are shown in indicated colors. Genes that showed characteristic responses are also shown. FDR: false discovery rate. *r*: Pearson correlation coefficient. (**B**) Gene groups that showed significant transcriptional changes. Among the KEGG orthology level 3 categories, gene groups in the category that were significantly up-regulated

*Figure 3 continued on next page*

*Figure 3 continued*

or down-regulated (FDR <0.05) in mox or mox-YG overexpression are shown in the violin plots. Genes with significant expression changes (FDR <0.05) within the same category are shown by swarm plots. Asterisks indicate significant differences (FDR <0.05) between mox and mox-YG comparisons in gene groups belonging to the same category. Comparisons for all categories are shown in *Figure 3—figure supplement 1C*. (**C**) Expression changes in representative amino acid, ammonium, and glucose transporters. (**D**) Verification of promoter activity by the reporter assay. Constructs used for promoter analysis with transcription reporters and quantitative results of RFP fluorescence values (arbitrary unit) for each promoter. 'moxFS' means the frameshift mutant of mox-YG is overexpressed. Bars and error bars indicate the mean and SD of maximum fluorescence values for RFP calculated from four biological replicates. Raw data are shown as dot plots. The p-values were calculated by performing Welch's *t*-test and applying Bonferroni correction. 'n.s.' means p>0.05. Detailed constructs and time series data for promoters other than those shown here are in *Figure 3—figure supplement 2*.

The online version of this article includes the following figure supplement(s) for figure 3:

**Figure supplement 1.** Comparison of the transcriptional response of mox and mox-YG overexpressing cells.

**Figure supplement 2.** Verification of promoter activity by the reporter assay.

**Figure supplement 3.** Depletion of leucine in the gTOW experiment does not induce the *GAP1* promoter.

The transcriptional responses in mox-YG overexpression obtained by RNA-seq were confirmed by transcriptional reporter analysis (*Figure 3D*, *Figure 3—figure supplement 2*). In this reporter analysis, we also tested overexpression of a frameshift mutant (moxFS) that imposes a burden on transcription but not on translation (*Figure 2—figure supplement 6C, D*). As a result, most of the responses observed under mox-YG overexpression were not observed under moxFS overexpression, suggesting that these transcriptional responses are likely caused by a burden on translation. The only exception was *TDH3pro*, whose activity decreased even under moxFS overexpression. Since *TDH3pro* is used for driving overexpression in this system, this result suggests that transcriptional competition may occur as a result of overexpression. Notably, in our gTOW experiments, leucine depletion is employed to induce overexpression; however, we consider it unlikely that leucine depletion per se is responsible for the induction of *GAP1*, because *GAP1* expression remained unchanged in the vector control even under gradual leucine depletion (*Figure 3—figure supplement 3*).

## Overexpression of mox-YG causes a proteomic response that partially overlaps with TORC1 inactivation

Protein burden is an overload on synthetic resources, and thus, when this overload occurs strongly, it is expected that there would be a reduction in the amount of proteins other than those being overexpressed due to competition for synthetic resources. Indeed, in cells overexpressing mox-YG, the amounts of proteins other than mox-YG were reduced (*Figure 4A*). Interestingly, the total amount of protein per cell remained almost constant, with the amount of other proteins decreasing as mox-YG was overexpressed. This trend was generally true for the overexpression of other FP (*Figure 4—figure supplement 1*). This suggests that there is a mechanism (or constraint) that maintains the overall cellular protein levels.

The maintenance of total cellular protein levels under overexpression conditions may be due to constraints such as intracellular space limitations or limitations in the availability of nutrients in the medium. To investigate whether nutrient limitation contributes to this phenomenon, we attempted overexpression of mox and mox-YG in YPD medium, which is considered to be more nutrient-rich than the SC medium. Since leucine depletion could not be used to drive overexpression in this context, we employed an alternative strategy using the aureobasidin A resistance marker (*AUR1d*) to increase plasmid copy number (*Figure 4—figure supplement 2A*). In this experiment, mox-YG was expressed at approximately 20% of the total cellular protein, with a corresponding increase in total protein content per cell (*Figure 4—figure supplement 2C, D*). This result suggests that in nutrient-rich YPD medium, cells may possess the capacity to buffer excess protein expression by increasing the total proteome. However, under these expression conditions, no growth inhibition was observed (*Figure 4—figure supplement 2B*), and thus the induction of mox-YG was not sufficiently strong. We therefore consider it necessary to investigate the regime under conditions of stronger overexpression in future studies.

To examine changes in protein expression, proteomic analysis of mox-YG overexpressing cells (in the SC medium) was performed. The distribution of intensities obtained from the analysis is shown in the Proteomaps in *Figure 4B*. In mox-YG overexpressing cells, mox-YG accounted for 40% of the

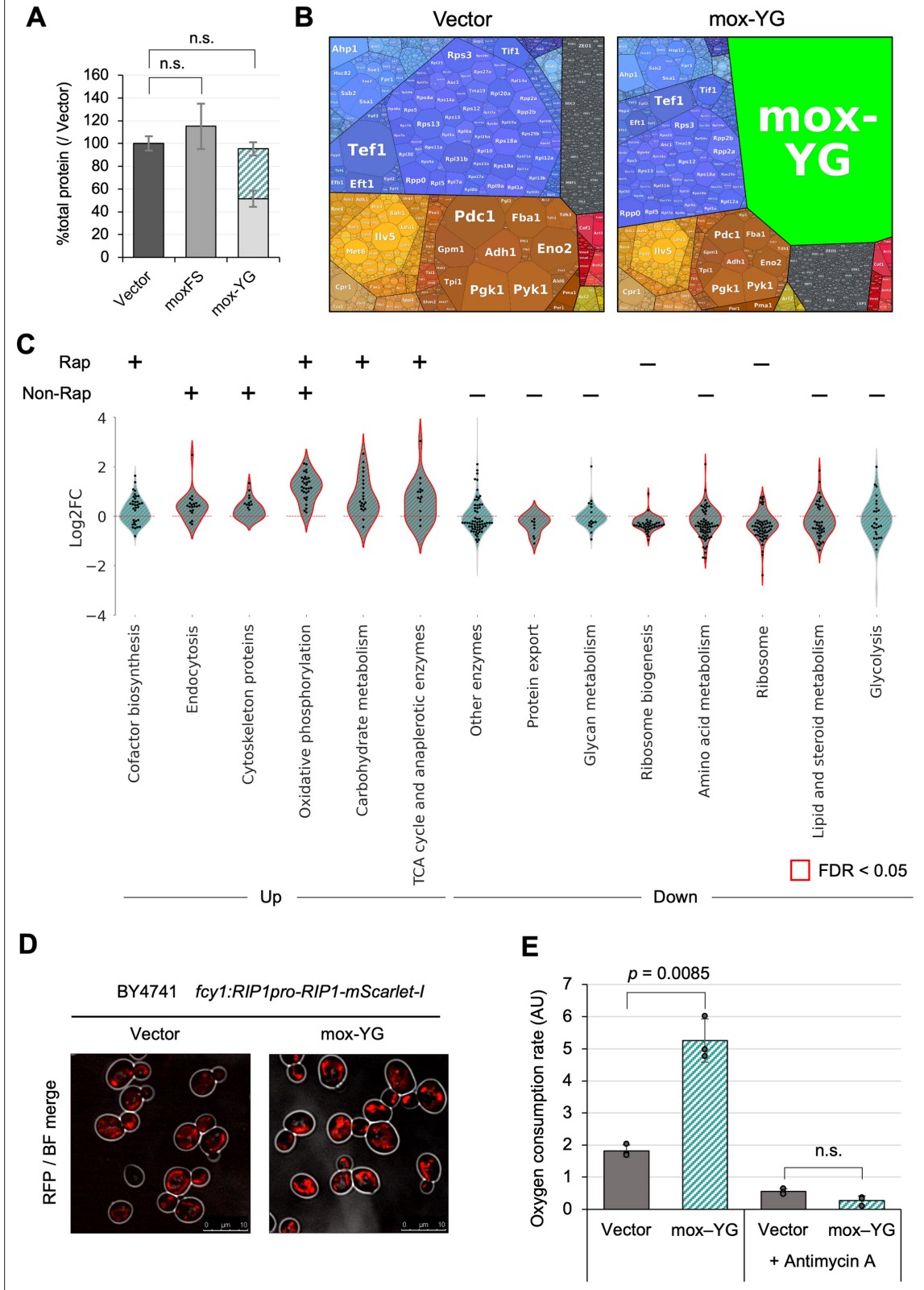

**Figure 4.** Overexpression of mox-YG causes a proteome response like that of TORC1 inactivation. (**A**) Ratio of the total protein level over the vector control. The colored area indicates the amount of overexpressed protein. The amount was calculated from the intensity of the target bands separated by SDS-PAGE. Bars and error bars indicate the mean and SD of protein levels calculated from three biological replicates. The p-values were calculated by performing Welch's *t*-test and applying Bonferroni correction. 'n.s.' means p>0.05. (**B**) Visualization of the proteome of the vector control and mox-YG

*Figure 4 continued on next page*

*Figure 4 continued*

overexpressing cells using Proteomaps (*Liebermeister et al., 2014*). Each polygon represents the mass fraction of the protein within the proteome. Similar colors indicate similar or closely related pathways and proteins. (**C**) Violin plots showing groups of genes in the KEGG orthology level 3 category whose expression was increased or decreased by mox-YG overexpression. Gene groups with significantly altered expression (FDR <0.05) in mox-YG overexpression are shown in red. Genes with significant expression changes (FDR <0.05) within the same category are shown by swarm plots. '+' and '−' indicate the categories with significantly increased or decreased expression, respectively, when separated into Rapamycin-responsive (Rap) and Rapamycin-non-responsive (Non-Rap) gene populations. Comparisons for all categories are shown in *Figure 4—figure supplement 6A, B*. Published data (*Gowans et al., 2018*) was used for comparison with the rapamycin response genes. (**D**) Fluorescence microscopy image showing the localization of Rip1-mScarlet-I. The vector control and mox-YG overexpressing cells were observed in the log phase. Stationary phase cells, bright-field images, quantitative results, and statistical analyses are shown in *Figure 4—figure supplement 8*. (**E**) Oxygen consumption rate in vector control and mox-YG overexpressing cells (arbitrary units, AU). Antimycin A was added as a control for respiratory inhibition. Measurements were conducted with three biological replicates. Bars and error bars indicate the mean and SD from three biological replicates. Raw data are shown as dot plots. The p-values were calculated by performing Welch's *t*-test and applying Bonferroni correction. 'n.s.' indicates p>0.05. The measurement data are shown in *Figure 4—figure supplement 9*.

The online version of this article includes the following source data and figure supplement(s) for figure 4:

**Figure supplement 1.** Change in total protein levels upon overexpression of target proteins.

**Figure supplement 2.** Evaluation of expression constraint of fluorescent proteins under nutrient-rich medium.

**Figure supplement 2—source data 1.** Original files for SDS-PAGE analysis displayed in *Figure 4—figure supplement 2C*.

**Figure supplement 3.** Comparison of normalized or not-normalized proteome results.

**Figure supplement 4.** Violin plots showing groups of proteins whose expression was changed by mox-YG overexpression.

**Figure supplement 5.** Cells overexpressing mox-YG exhibit proteomic changes that overlap with those observed in rapamycin-treated cells.

**Figure supplement 6.** Comparison of groups of proteins whose expression levels are changed by mox-YG overexpression and rapamycin treatment.

**Figure supplement 7.** Assessment of TORC1 pathway activity in mox-YG overexpressing cells.

**Figure supplement 7—source data 1.** Original files for SDS-PAGE analysis displayed in *Figure 4—figure supplement 7F*.

**Figure supplement 8.** Observation of mitochondria in mox-YG overexpressing cells.

**Figure supplement 9.** Time-course analysis of oxygen consumption in mox-YG overexpressing cells.

total protein expression. *Figure 4A* suggests that in mox-YG overexpressing cells, proteins other than mox-YG are overall reduced to about 60% compared to the vector. Therefore, we considered it necessary to normalize the data overall to accurately represent changes in the per-cell proteome. However, in this study, we used data without such correction to focus on how the composition of the proteome, other than mox-YG, changed (see Materials and methods and *Figure 4—figure supplement 3A–C* for more details).

*Figure 4C* illustrates the categories that showed significant changes upon mox-YG overexpression (all categories are shown in *Figure 4—figure supplement 4*). Elevated mitochondrial proteins, such as oxidative phosphorylation and TCA cycle, and decreased ribosomal protein and ribosomal synthesis were observed. Interestingly, the proteomic changes observed in mox-YG overexpressing cells showed little correlation with the corresponding transcriptomic changes (*Figure 4—figure supplement 3D*). The only notable exception was oxidative phosphorylation, which was upregulated at both the transcript and protein levels (*Figures 3B and 4C*). In contrast, for example, ribosomal proteins were upregulated at the transcript level but markedly downregulated at the protein level (*Figures 3B and 4C*). This discrepancy may indicate a cellular state in which transcriptional responses are uncoupled from protein expression as a result of protein burden.

As mentioned above, RNA-seq results suggested nitrogen source starvation in mox-YG overexpression. Therefore, we investigated whether this response is influenced by the regulation through the TORC1 pathway, a primary mechanism responsive to nitrogen sources. By comparing expression changes due to rapamycin (a TORC1 inhibitor) treatment (*Gowans et al., 2018*) and mox-YG overexpression, we found that about 40% of the proteins with altered expression were congruent with the group of proteins regulated by TORC1 (*Figure 4—figure supplement 5A, B*). Thus, the proteome expression changes due to mox-YG overexpression may include responses due to inactivation of the TORC1 pathway as well as other responses. *Figure 4C* shows which categories had significant expression changes when separated into rapamycin-responsive (Rap) and rapamycin-non-responsive (Non-Rap) gene populations. Carbohydrate metabolism, elevated TCA cycle, and decreased ribosome synthesis were suggested to be responses due to TORC1 inactivation, while endocytosis, cytoskeleton,

protein transport and lipid synthesis, and glycolysis were suggested to be due to responses other than TORC1 inactivation (*Figure 4—figure supplement 6*). Oxidative phosphorylation included both Rap responses (12/32) and non-responses (20/32) (*Figure 4—figure supplement 6C*). Indeed, mox-YG overexpression reduced the sensitivity to rapamycin compared to the vector control (*Figure 4—figure supplement 7A–E*), supporting the notion that TORC1 activity was diminished. However, we were unable to obtain direct evidence demonstrating TORC1 inactivation. We examined the phosphorylation status of Atg13 (*Kamada et al., 2000*), a known TORC1 target, by Western blotting. In mox-YG overexpressing cells, however, the total expression level of Atg13 was markedly reduced (*Figure 4—figure supplement 7F*). Therefore, it remains inconclusive how the decreased expression of TORC1 targets affects downstream signaling under these conditions.

Above transcriptome and proteome analyses suggested that mox-YG overexpression induces mitochondrial development and a metabolic shift from fermentation to oxidative respiration. To verify mitochondrial development, we performed fluorescence microscopy. Indeed, mitochondria labeled with Rip1-mScarlet-I, a component of the oxidative phosphorylation pathway, were significantly more developed in mox-YG overexpressing cells (*Figure 4D*, *Figure 4—figure supplement 8*). Furthermore, oxygen consumption was increased in these cells (*Figure 4E*, *Figure 4—figure supplement 9*), supporting the notion that they had undergone a metabolic shift toward oxidative respiration.

## Overexpression of mox-YG causes a dysplastic nucleolus, which is alleviated by mutations in the nuclear exosome

Ribosomes are a crucial factor for protein burden because they are themselves protein synthesizers while being synthesized in the highest amounts in the cell. Therefore, we further focused on the behavior of ribosomes in mox-YG cells, and expression changes were investigated in detail for each ribosomal protein (*Figure 5—figure supplement 1*). While the majority of the 84 ribosomal proteins detected (45 of them significantly) decreased, six (Rps12, Rps31, Ppp0, Rpp2a, Rpl5, and Rpl12) showed significant increases. Interestingly, all decreased ribosomal proteins are assembled into ribosomes in the nucleolus, while all increased ribosomal proteins are assembled into ribosomes in the cytoplasm (*Lafontaine, 2015*). They constitute the 'beak' of the 40 S subunit and the 'P-stalk' and the 'central protuberance (pc)' of the 60 S subunit (*Figure 5A*). Electron micrographs showed an abnormal nucleolar formation of mox-YG overexpressing cells (*Figure 5B*, *Figure 5—figure supplement 3*), in addition to reduced ribosome density in the cytoplasm (*Figure 5—figure supplement 2A*). This abnormal nucleolar formation was also confirmed by fluorescence microscopy images of Nsr1 (Nucleolin), the major protein of the nucleolus (*Figure 5—figure supplement 2B–D*).

Based on these results, we hypothesized that overexpression of mox-YG creates an imbalance between the amount of ribosomal protein synthesized in the cytoplasm and the amount of rRNA synthesized in the nucleolus (*Figure 5C and D*). The imbalance may lead to the creation of misassembled ribosomes, which are eventually degraded. We speculated that the primary enzyme degrading the misassembled ribosome should be the nuclear exosome. This is because our previous genome-wide mutation analysis identified exosome mutants (*dis3-1, mtr3-ts, mtr4-1, rpn42-ph, rpn45-ph, and ski6-ph*) as primary mitigators of protein burden (*Kintaka et al., 2020*). We confirmed that the *mtr4-1* mutant does not show growth retardation upon mox-YG overexpression (*Figure 5E*). In fact, in *mtr4-1* cells, there was no difference in growth rate between the vector control and mox-YG overexpression (*Figure 5—figure supplement 5B*), but rather a shorter lag time for the mox-YG overexpression than the vector control (*Figure 5E*). We next quantified protein upon mox-YG overexpression in *mtr4-1* cells (*Figure 5F*). Compared to the wild-type (Vector), the total protein content in *mtr4-1* (Vector) cells was reduced to 78%, which was likely related to the decreased growth. In the *mtr4-1* mox-YG overexpressing cells, the amount of other proteins decreased (to 54%), overwhelmed by the 40% expression of mox-YG. However, the overall protein content increased compared to the *mtr4-1* vector control, with a composition like the wild-type mox-YG overexpressing cells. Therefore, in *mtr4-1* cells, even though mox-YG overexpression reduced the amount of other proteins (to 54%) compared to the vector control (78%), the growth rate was not reduced (*Figure 5—figure supplement 5B*). This suggests that mox-YG overexpression restores the effect of growth inhibition caused by *mtr4-1*. Next, we observed the composition of the nucleolus in *mtr4-1* (*Figure 5G*, *Figure 5—figure supplement 5D, E*). The failure of nucleolus formation observed upon mox-YG overexpression was partially, but significantly, rescued in the *mtr4-1* strain. This supports our hypothesis that misassembled ribosomes

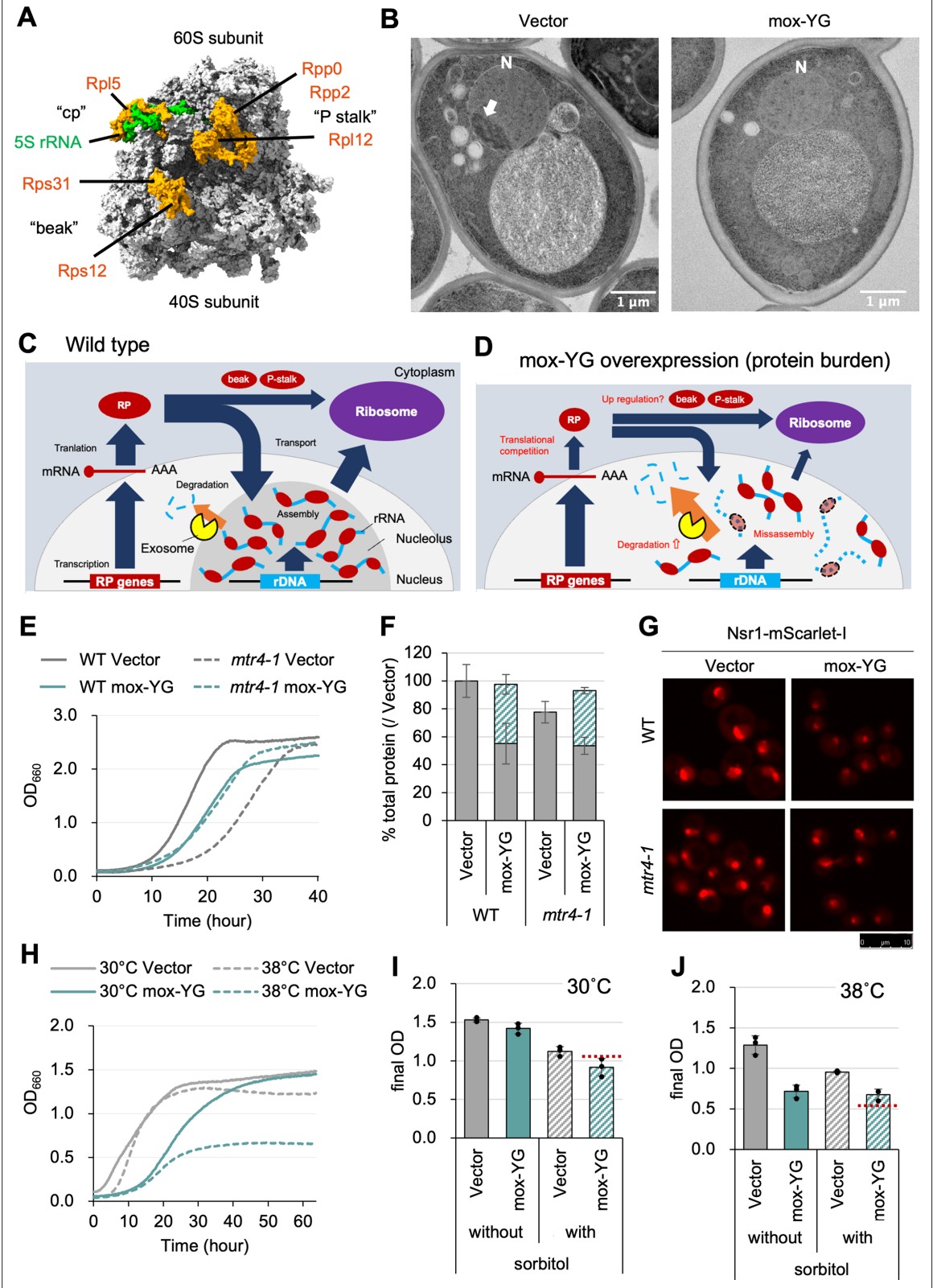

**Figure 5.** mox-YG overexpression causes abnormal nucleolus formation. (**A**) 3D structure of the ribosome. PDB model 6GQV (*Pellegrino et al., 2018*) is used for the base model and colored using ChimeraX software (*Meng et al., 2023*). Ribosomal proteins with increased expression upon mox-YG overexpression are shown with the named structures containing them. See also *Figure 5—figure supplement 1* for the quantitative data. (**B**) Electron microscope images of the vector control and mox-YG overexpressing cells. The arrow in the image points to the nucleolus structure. N: nucleus. Images

*Figure 5 continued on next page*

*Figure 5 continued*

of other observed cells are shown in *Figure 5—figure supplement 3*. (**C, D**) Model diagrams showing the hypothetical situation of the wild type (**C**) and mox-YG overexpressing cells (**D**). In WT cells, enough ribosomal proteins (RPs) are produced, resulting in RP-assembled rRNAs and the formation of nucleolus with normal morphology. On the other hand, in mox-YG overexpressing cells, the amount of RP is reduced due to translation competition, increasing misassembled rRNA. As a result, degradation of rRNA by exosomes may be accelerated, resulting in abnormal nucleolus morphology. (**E**) Growth curves of WT and *mtr4-1* mutant cells with the vector control or upon overexpression of mox-YG at 30 °C. The solid or dotted lines show the average calculated from three biological replicates. Growth curves with SDs of replicates are shown in *Figure 5—figure supplement 5A*. (**F**) Ratio of total protein levels of WT and *mtr4-1* strains with the vector or upon mox-YG overexpression, calculated the total protein level of WT cells with the vector as 100%. Gray bars indicate expression levels of proteins other than mox-YG; green shaded bars indicate mox-YG expression levels. Error bars were SDs calculated from three biological replicates. The p-values were calculated by performing Welch's *t*-test. 'n.s.' means p>0.05. (**G**) Fluorescence microscopy image of nucleolus-localized protein Nsr1-mScarlet-I of the WT and *mtr4-1* mutant cells with the vector or mox-YG overexpression. Bright field and merged images, and quantification of the nucleolus size are shown in *Figure 5—figure supplement 5D and E*. (**H**) Growth curves of the cells with the vector or under mox-YG overexpression cultivated at 30 °C or 38 °C. The solid or dotted lines show the average calculated from three biological replicates. Growth curves with SDs of replicates are shown in *Figure 5—figure supplement 6A*. (**I, J**) Final OD of the cell culture with or without 1 M sorbitol at 30 °C (**I**) or 38 °C (**J**). The bars and error bars show the means and SDs calculated from three biological replicates. The red dotted line indicates the final OD estimated from the product of (mox-YG without sorbitol) / (Vector without sorbitol) and (Vector with sorbitol) / (Vector without sorbitol). Growth curves with SDs of replicates are shown in *Figure 5—figure supplement 6C, D*.

The online version of this article includes the following source data and figure supplement(s) for figure 5:

**Figure supplement 1.** Expression changes of ribosomal proteins upon mox-YG overexpression.

**Figure supplement 2.** Microscopic analysis of the cells under mox-YG overexpression.

**Figure supplement 3.** Electron microscopic images of the cells with the vector (**A**) or under mox-YG overexpression (**B**).

**Figure supplement 4.** Model diagrams showing the hypothetical situation of the cells.

**Figure supplement 5.** Positive genetic interaction between exosome mutation (*mtr4-1*) and mox-YG overexpression.

**Figure supplement 5—source data 1.** Original files for SDS-PAGE analysis displayed in *Figure 5—figure supplement 5C*.

**Figure supplement 6.** Comparison of growth of glycolytic enzymes overexpressing strains.

are degraded by the exosome (*Figure 5D*), and with low exosome activity, nucleolus formation (and efficient ribosome assembly) would be possible even with overexpression of mox-YG (*Figure 5—figure supplement 4D*).

While experimenting with mox-YG overexpressing cells, we found that these cells stop growth earlier (lower final density) at higher temperatures (38 °C) than the vector controls (*Figure 5H*, *Figure 5—figure supplement 6A*). Cells at this time did not show cell death such as cell bursting (*Figure 5—figure supplement 6B*). Recently, it was reported that intracellular osmotic pressure increases at high temperatures, resulting in less formation of liquid-liquid phase-separated structures such as nucleoli, which are recovered when cells are placed in hyperosmotic conditions (*Watson et al., 2023*). We hypothesized that the growth inhibition of mox-YG overexpressing cells at high temperatures may be due to the enhanced failure of nucleolus formation, and thus cultured mox-YG overexpressing cells under hyperosmotic pressure. At 30 °C, hyperosmolarity caused the same decrease in growth for the vector control and mox-YG overexpression (*Figure 5I*, *Figure 5—figure supplement 6C*). On the other hand, at 38 °C, hyperosmolarity caused a decrease in growth only in the vector control and no further decrease in mox-YG overexpression cells (*Figure 5J*, *Figure 5—figure supplement 6D*). The same trend was observed regarding the size of the nucleolus (*Figure 5—figure supplement 6E, F*). Under the conditions with added sorbitol and under mox-YG overexpression, the size of the nucleoli was smaller than the control at 30 °C, whereas, at 38 °C, it was rather larger than the control. This suggests that mox-YG overexpression and hyperosmolarity may mitigate each other's growth-inhibitory effects, and one explanation may be that hyperosmolarity suppresses the abnormal organization of the nucleolus caused by mox-YG overexpression.

## The Gpm1 mutant shows a similar NI to mox-YG, yet yields distinct phenotypes upon overexpression

Thus far, our investigation has focused on heterologous fluorescent proteins. Finally, we turned to endogenous yeast proteins to identify those with the lowest cytotoxicity and to examine the phenotypic consequences of their overexpression. We previously investigated the expression limits of a group of glycolytic enzymes in *S. cerevisiae*, with Gpm1 and its catalytic center mutant (CCmut) having

the highest expression limits (*Eguchi et al., 2018*). Here, we evaluated the neutrality of the wild types and their CC-mutants of Tdh3 and Gpm1 (*Figure 6A*, *Figure 6—figure supplement 1*). The growth rates at the expression limits were higher in the CCmut for both Tdh3 and Gpm1, and in Gpm1, the CCmut grew twofold faster (*Figure 6—figure supplement 1A, C*). The protein levels of wild types and mutants of both proteins were not significantly different (*Figure 6—figure supplement 1D, E*). The calculated NIs were highest for Gpm1–CCmut, which was equivalent to that of mox-YG (*Figure 6A*). We previously concluded that the activities of glycolytic enzymes generally do not affect their expression limits (*Eguchi et al., 2018*), but their neutralities suggest that their activities can become constraints in some way. This may be due to a disturbance in metabolism or a property of the active enzyme, such as an effect of reactive amino acids (His or Cys, *Figure 6—figure supplement 2*).

Although Gpm1 and Gpm1–CCmut exhibited NIs comparable to that of mox-YG, suggesting that they are similarly unconstrained, notable differences in cell morphology were observed (*Figure 6B*, *Figure 6—figure supplement 3*). Cells overexpressing Gpm1 appeared enlarged and spherical, while those overexpressing Gpm1–CCmut did not show such hypertrophy but were still rounder than vector controls or mox-YG overexpressing cells. The transcriptional responses observed in mox-YG over-expressing cells, including the induction of *GAP1*, a marker of the amino acid starvation response, were not detected in Gpm1–CCmut overexpression (*Figure 6C*, *Figure 6—figure supplement 4*). In Gpm1–CCmut overexpressing cells, similar to mox-YG overexpression, the levels of proteins other than Gpm1–CCmut were reduced, despite a slight, statistically non-significant increase in total protein amount (*Figure 6—figure supplement 1F*). While mitochondrial development was observed in Gpm1–CCmut cells using Rip1–mScarlet-I labeling (*Figure 6—figure supplement 5*), no increase in oxygen consumption was detected (*Figure 6D*, *Figure 6—figure supplement 6*). The nucleolus, visu-alized with Nsr1–mScarlet-I, appeared similarly shrunken in Gpm1–CCmut overexpressing cells as in mox-YG overexpressing cells (*Figure 6E*, *Figure 6—figure supplement 7*). These results suggest that the phenotypes observed in mox-YG overexpressing cells include both mox-YG-specific effects and responses commonly associated with protein burden. Alternatively, as indicated by the abnormal cell morphology, Gpm1–CCmut may possess unknown constraints or cytotoxic properties not captured by the neutrality index, which could obscure the detection of phenotypes typically associated with protein burden.

## Discussion

In this study, we found that the fluorescent properties of FPs constrain their expression limits. This fluorescence-induced cytotoxicity appears as different phenotypes depending on the FPs; an excess of fluorescent mCherry causes cell enlargement, but such a phenotype was not seen with green fluo-rescent proteins. On the other hand, the cause of the cytotoxicity is not clear currently. Fluorescence itself does not seem to constrain the expression limit, because the mox expression limit remained the same when cultured in the dark or under strong light (*Figure 2—figure supplement 7*). We also believe that $H_2O_2$ generated during fluorophore maturation (*Craggs, 2009*) is unlikely. In the presence of $H_2O_2$, mox overexpressing cells grew slightly slower than mox-YG overexpressing cells (*Figure 2—figure supplement 8H*). However, no transcriptional response to oxidative stress, including the cata-lase gene (*CTT1*), was seen with mox overexpression (FDR <0.187, over the vector control, Data S1). Because *CTT1* expression was elevated in mox-YG overexpression (FDR <0.006, over the vector control, Data S1), presumably due to enhanced respiration, mox-YG overexpression might rather increase oxidative stress tolerance. Conversely, 'inactive' proteins might accumulate more without interfering with intracellular functions because both inactive fluorescent proteins and inactive glyco-lytic enzymes showed higher neutrality (*Figures 2 and 6*). However, our current data did not clarify the mechanism, and investigating more proteins and their variants would be required.

The currently least cytotoxic proteins, non-fluorescent mox-YG and inactive Gpm1, express more than 40% of the total protein, whereas yeast cells can maintain growth under these conditions (*Figures 2 and 6*). This is higher than the prediction in *E. coli*, which stops growing when 37% of the excess protein is expressed (*Bruggeman et al., 2020*). This may be due to the very low cytotoxicity of the proteins expressed in this study, or it may be because eukaryotic cells have greater capacity than prokaryotic cells. In the mox-YG overexpressing cells, we observed a previously expected but unobserved cellular physiology of the protein burden: depletion of protein synthesis resources (i.e. nitrogen source; *Figure 3*) and a decrease in other proteins (glycolytic enzymes and ribosomes;

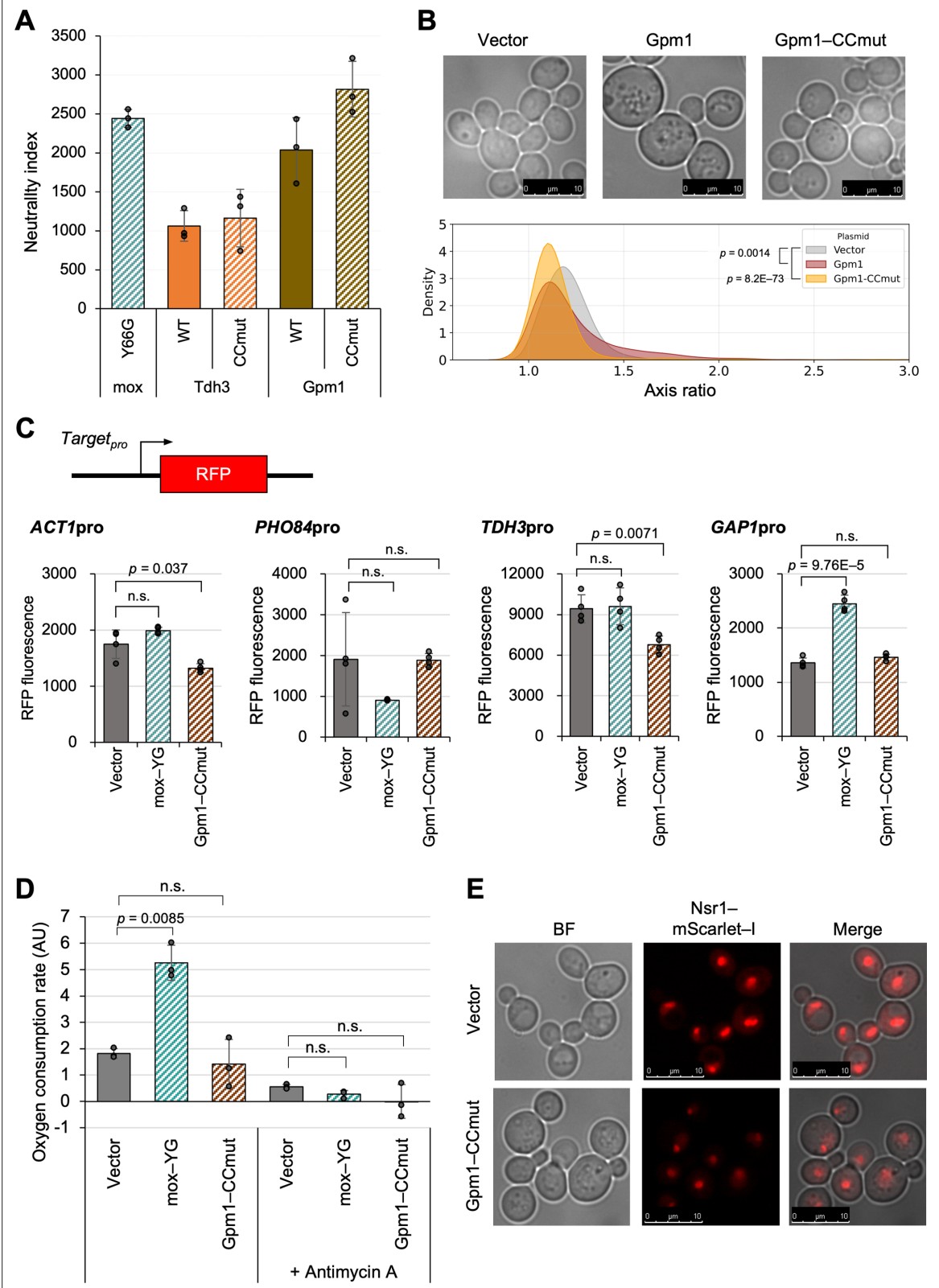

**Figure 6.** The Gpm1 mutant shows a similar NI to mox-YG, yet yields distinct phenotypes upon overexpression. (**A**) The neutrality index for Gpm1, Tdh3, and their CCmuts. Data on growth rates and protein expression levels used for the neutrality index calculation are presented in *Figure 6—figure supplement 1*. (**B**) Top images: representative microscopic images of cells overexpressing Gpm1 or Gpm1–CCmut. Bottom graph: Distribution of cell axis ratios upon overexpression of Gpm1 and Gpm1–CCmut, compared with the vector control. The p-values were calculated by the Mann–Whitney

*Figure 6 continued on next page*

Figure 6 continued

*U*-test. Statistical analysis of cell size and comparison with mox-YG overexpressing cells are shown in *Figure 6—figure supplement 3*. (C) Promoter activity reporter assay. Constructs used for promoter analysis with transcription reporters and quantitative results of RFP fluorescence values (arbitrary unit) for each promoter. Time series data are in *Figure 6—figure supplement 4*. (D) Oxygen consumption rate in vector control, mox-YG, and Gpm1-CCmut overexpressing cells (arbitrary units, AU). Antimycin A was added as a control for respiratory inhibition. The measurement data are shown in *Figure 6—figure supplement 6*. (E) Fluorescence microscopy images of nucleolus with Nsr1–mScarlet-I. Log phase cells (OD$_{660}$=1.0) with the control vector or under Gpm1 or Gpm1–CCmut overexpression were observed. Statistical analysis of nucleolar size is shown in *Figure 6—figure supplement 7*. The bars and error bars in **A**, **C** and **D** show the means and SDs calculated from three biological replicates. The raw data is shown with dot plots. The p-values were calculated by performing Welch's *t*-test and applying Bonferroni correction. 'n.s.' means p>0.05.

The online version of this article includes the following source data and figure supplement(s) for figure 6:

**Figure supplement 1.** Evaluation of the neutrality of Tdh3 and Gpm1.

**Figure supplement 1—source data 1.** Original files for SDS-PAGE analysis displayed in *Figure 6—figure supplement 1D*.

**Figure supplement 2.** Nucleotide and amino acid sequences of Gpm1 and Tdh3.

**Figure supplement 3.** Overexpression of Gpm1 and its CC-mutant induces abnormal cell morphology.

**Figure supplement 4.** Transcriptional response under mox-YG and Gpm1 CC-mutant overexpression.

**Figure supplement 5.** Microscopic observation of mitochondria in cells overexpressing Gpm1 or Gpm1–Ccmut.

**Figure supplement 6.** Time-course analysis of oxygen consumption in Gpm1–CCmut overexpressing cells.

**Figure supplement 7.** Microscopic observation of nucleoli in cells overexpressing Gpm1 or Gpm1–Ccmut.

*Figure 4*), possibly due to competition for synthesis. In addition, metabolic conversion to mitochondria and failure of nucleolus formation were observed (*Figures 4 and 5*). As a matter of fact, the nitrogen source depletion response (*Figure 3*) itself is not self-evident. Because the total protein content in the mox-YG overexpressing cells is the same as that in the vector control (*Figure 4A*), the intracellular nitrogen source used for protein synthesis should not be depleted. Then, why does the nitrogen source depletion response occur? Excess mox-YG may promote wasteful consumption of amino acids and nitrogen, or there may be pathways that sense the amount of 'useful proteins' and their ability to be synthesized, or there may be depletion of specific amino acids in mox-YG excess. Another possibility is that there may be some mechanism that maintains the amount of total protein in the cell even when the nitrogen source is depleted. Depletion of nitrogen sources probably results in the inactivation of the TORC1 pathway (*Figure 4C*), metabolic conversion from fermentation to respiration, and ribosome depression to an economic mode. In *E. coli*, the ppGpp pathway plays this role (*Shachrai et al., 2010*). While the above is thought to be a physiological response to protein burden, a defect in ribosome synthesis in the nucleolus was also suggested (*Figure 5B*). This could be due to a conflict between nuclear rRNA synthesis and cytoplasmic protein synthesis or to the negative effects of the inability to create a liquid-liquid phase-separated structure due to dilution effects (see below). The partial restoration of nucleolus formation by a mutation of the exosome (*Figure 5G*) supports this possibility.

When one type of protein is expressed as much as 40% of the total, as in the present study, the cytoplasmic crowding, viscosity, pH, etc. may change due to the nature of the protein and the effect of the reduced protein by competition. These may alter the movement of substances. It is known that when ribosomes are reduced, the fluidity of large proteins changes (*Delarue et al., 2018*). Water activity-osmotic pressure may change (*Watson et al., 2023*), and pH may also be affected. These may affect the formation of liquid-liquid phase-separated structures (*Delarue et al., 2018*; *Watson et al., 2023*). The failure of nucleolus formation observed in mox-YG overexpression (*Figure 5B*) may be due to such a condition change. In fact, the growth of mox-YG overexpressing cells is significantly impaired at high temperatures (*Figure 5H*), where liquid-liquid phase separation is less likely to occur (*Watson et al., 2023*). In contrast, the effect appears to be reduced at high osmolarity (*Figure 5J*), where it is more likely to occur (*Watson et al., 2023*). A mutation of the nuclear exosome restored nucleolus formation (*Figure 5G*), which might be due to the reformation of the liquid-liquid phase-separated structure (i.e. nucleolus) by increasing the concentration of rRNA. These have not been considered phenomena that occur with protein overexpression. The engineering of physicochemical properties of the cytoplasm thus may be possible by mass expression of proteins with specific properties.

In this study, in addition to mox-YG, inactive Gpm1 was also expressed in large amounts as well (*Figure 6*). Both are globular proteins with relatively small molecular weight, but it is not yet clear what

characteristics are necessary for them not to be cytotoxic. Further investigation of the neutrality of various proteins will clarify these features. In addition, while the overexpression of Gpm1–CCmut also caused nucleolar shrinkage similar to that observed in mox-YG overexpressing cells, it did not induce other phenotypes such as nitrogen starvation or enhanced respiration (*Figure 6*). Therefore, it remains unclear which of the phenotypes observed in mox-YG overexpressing cells are general consequences of 'protein burden' and which are specific to mox-YG. As noted in the Introduction (*Figure 1E*), it is essential to continue investigating multiple proteins that appear to be 'unconstrained' in order to address this issue. In this study, expression limits were investigated, but no experiments were performed to gradually increase expression levels. For example, it would be meaningful to investigate at what expression level of mox-YG a shift to the economy mode or failure of nucleolus formation occurs. Also, due to the limitations of the experimental system, it may not be possible to express mox-YG to its true limit. The expression of one specific protein cannot be 100% of the total protein, and there is always a limit. As the number of ribosomes continues to decrease due to competition for synthesis, the excess proteins cannot be synthesized either. Thus, there must be a final equilibrium point. A more powerful and controllable expression system would be needed to investigate this. Although the translation system appears to be the primary target of burden in mox-YG overexpression (*Figure 2—figure supplement 6C, D*), the bottleneck in the expression system may also differ depending on gene design (*Figure 1D*; *Hausser et al., 2019*).

## Materials and methods

### Strains, growth conditions, and yeast transformation

BY4741 (*MATa his3Δ1 leu2Δ0 met15Δ0 ura3Δ0*) (*Brachmann et al., 1998*) was used as the host strain for the experiments. Yeast culture and transformation were performed as previously described (*Amberg et al., 2005*). A synthetic complete (SC) medium without uracil (Ura) or leucine (Leu), as indicated, was used for yeast culture. Other strains used in this study are listed in the Key Resource Table.

### Plasmid construction

The plasmids were constructed by homologous recombination activity of yeast cells (*Oldenburg et al., 1997*), and their sequences were verified by DNA sequencing. Plasmids used in this study are listed in the Key Resource Table.

### Genetic tug-of-war

To overexpress a target protein to a level that causes growth inhibition (in this study, this is referred to as 'expression limit'), we used gTOW (*Figure 2—figure supplement 1A*; *Moriya et al., 2012*; *Moriya et al., 2006*). The gene of the target protein is incorporated into the gTOW plasmid (here, pTOW40836) and introduced into yeast cells lacking *ura3* and *leu2* genes. The selection marker in this case is uracil (–Ura conditions). Since this plasmid has the replication origin of 2 μm plasmid (*2μ ORI*), this plasmid becomes multicopy in the cell (about 30 in the case of the vector). When yeast cells carrying the plasmid are transferred to a medium without leucine and uracil (–Leu/Ura conditions), the copy number of the plasmid in the cells increases (to about 120 in the case of the vector). This is because *leu2-89* on the plasmid is a *LEU2* allele with a large deletion in its promoter, so cells with higher plasmid copy numbers will grow faster in –Leu/Ura conditions. In other words, *leu2-89* acts as a bias to raise the plasmid copy number in –Leu/Ura conditions. If overexpression of the target protein inhibits growth, the plasmid copy number will be lower than the copy number that gives rise to the critical expression of that protein. In other words, the target gene acts as a bias to lower the plasmid copy number. The resulting tug-of-war between the selection biases created by the two genes leads to a target protein being expressed at close to the critical expression level in –Leu/Ura conditions (the green-colored area in *Figure 2—figure supplement 1A*). By using a version of the aureobasidin A resistance gene (*AUR1*) with a deletion in its promoter (*AUR1d*) as a bias to increase plasmid copy number, experiments based on the same concept as gTOW can also be conducted in rich medium (YPD + 500 ng/mL aureobasidin A). In this case, the plasmid pTOW-AUR1d is used (*Figure 4—figure supplement 2A*).

### Measuring growth rate

Cellular growth was measured by monitoring $OD_{660}$ every 10 min using a compact rocking incubator (TVS062CA, ADVANTEC). The max growth rate (MGR) was calculated as described previously (*Moriya*

et al., 2006). Average values, SD, and p-values of Welch's *t*-test were calculated from at least three biological replicates. The growth data of *Figure 2—figure supplement 10A* was measured by monitoring $OD_{595}$, respectively, every 10 min using a microplate reader (Infinite F200, TECAN).

## Protein analysis

Yeast cells overexpressing a target protein were cultivated in SC–Leu/Ura medium. The total protein was extracted from log-phase cells ($OD_{660}$ = 1.0) with a NuPAGE LDS sample buffer (Thermo Fisher Scientific) after 0.2 mol/l NaOH treatment (*Kushnirov, 2000*). For each analysis, the total protein extracted from 1 optical density (OD) unit of cells with $OD_{660}$ was used. For total protein visualization, the extracted total protein was labeled with Ezlabel FluoroNeo (ATTO), as described in the manufacturer's protocol, and separated by 4–12% sodium-dodecyl sulfate acrylamide gel electrophoresis (SDS-PAGE). Proteins were detected and measured using the LAS-4000 image analyzer (GE Healthcare) in SYBR–green fluorescence detection mode and Image Quant TL software (GE Healthcare). Protein quantification was performed as shown in *Figure 2—figure supplement 1B*. Average values, SD, and p-values of Welch's *t*-test were calculated from three biological replicates.

## Western blotting of Atg13

Cells overexpressing FPs were collected at $OD_{660}$=1 or 2 and fixed with 100 µL of ice-cold alkaline solution (0.2 N NaOH and 0.5% β-mercaptoethanol). After 5 min of incubation on ice, 1 mL of ice-cold acetone was added to the sample and incubated at −20 °C to precipitate proteins. The protein samples were precipitated with a microfuge (15,000 rpm, 5 min), air-dried, suspended in 100 µL of SDS–PAGE sample buffer, and incubated at 65 °C for 15 min. The samples were thoroughly dissolved by sonication and subjected to SDS–PAGE (40 mA, 1 hr). After ponceau S staining, proteins are transferred to the PVDF membrane (Millipore #IPVH304F0) and blocked in 5% skim milk/TBST. Atg13 was detected using a rabbit anti-Atg13 antibody (1:3000) and peroxidase-conjugated goat anti-rabbit secondary antibody (Jackson ImmunoResearch, #111–035–003, 1:10,000). Chemiluminescent substrates (Millipore, #WBLUF0100) and Light–Capture II (ATTO) were used for signal detection.

## Measuring GFP fluorescence

GFP fluorescence (Ex485nm/Em535nm) was measured every 10 min using a microplate reader (Infinite F200, TECAN). Average values, SD, and *p*-values of Welch's *t*-test were calculated from eight biological replicates.

## RNA-seq analysis

BY4741 cells overexpressing a target protein were cultured in SC–Leu/Ura medium and harvested at the logarithmic growth phase. RNA extraction was performed according to *Köhrer and Domdey, 1991*. The purified RNA was quality checked by BioAnalyzer (Agilent) or MultiNA (Shimazu), and the concentration was measured by Qubit (Thermo Fisher Scientific). Purified RNA was stored at −80 °C until subsequent experiments. cDNA libraries were prepared using the TruSeq Stranded Total RNA kit (Illumina), using half the protocol of the TruSeq RNA library prep kit. Three biological replicates were analyzed for all strains. The sequences were checked for read quality by FastP (*Chen et al., 2018*) and then aligned using Hisat2 (*Kim et al., 2019*). The aligned data were formatted into bam files by Samtools (*Li et al., 2009*) and quantified by StringTie (*Pertea et al., 2015*). Finally, expression level variation analysis was performed by EdgeR (*Robinson et al., 2010*). The raw data were deposited into DDBJ (accession number: PRJDB18064). The processed data (*Source data 1*) included the average expression levels (log10CPM) of 6685 genes in the vector control, the changes in expression (log2FC) upon overexpression of mox or mox-YG compared to the vector control, and the significance values of expression change (mox_FDR, mox-YG_FDR). *Figure 3—figure supplement 1A, B* is the visualization of this data using volcano plots. This data is used in subsequent data analyses below.

## Reporter assay

The strains used for the reporter assay were created by introducing the promoter region of each gene and the sequence of mScarlet-I, a type of RFP, into the *FCY1* locus of the genome as shown in *Figure 3—figure supplement 2A*. Correct integration was checked by histidine prototrophy and 5-FC resistance of the strain. The cell density ($OD_{595}$) and RFP fluorescence (Ex535nm/Em590nm) of

the created strains were measured every 10 min using a microplate reader (Infinite F200PRO, TECAN). Average and SD of max RFP fluorescence, and *p*-values of Welch's *t*-test were calculated from eight biological replicates.

## Proteome analysis

The vector control and mox-YG overexpressing cells (three biological replicates) were cultured in 5 mL of SC–LeuUra medium with shaking, and cells were collected at $OD_{660}$=1. Cells were washed twice with PBS, frozen on dry ice, and transported to Kazusa DNA Research Institute. The samples were prepared as described in the previous study (*Hughes et al., 2019*; *Kawashima et al., 2022*). The cells were dissolved in 100 mM Tris-HCl (pH 8.0) containing 4% SDS and 20 mM NaCl using BIORUPTOR BR-II (SONIC BIO Co., Kanagawa, Japan). The 20 µg of extracted proteins were quantified using Pierce BCA Protein Assay Kit (Thermo Fisher Scientific, WA, USA) at 500 ng/µL. The protein extracts were reduced with 20 mM tris (2-carboxyethyl) phosphine for 10 min at 80 °C followed by alkylation with 30 mM iodoacetamide for 30 min at room temperature in the dark. Protein purification and digestion were performed using the sample preparation (SP3) method (*Kawashima et al., 2022*). The tryptic digestion was performed using 500 ng/µL Trypsin/Lys-C Mix (Promega, Madison, WI, USA) for overnight at 37 °C. The digests were purified using GL-Tip SDB (GL Sciences, Tokyo, Japan) according to the manufacturer's protocol. The peptides were dissolved again in 2% ACN containing 0.1% TFA and quantified using BCA assay at 150 ng/µL. The digested peptides were loaded directly using a 75 µm×12 cm nanoLC nano-capillary column (Nikkyo Technos Co., Ltd., Tokyo, Japan) at 40 °C and then separated with a 30 min gradient (mobile phase A=0.1% FA in water, B=0.1% FA in 80% ACN) consisting of 0 min 8% B, 30 min 70% B at a flow rate of 200 nL/min using an UltiMate 3000 RSLC-nano LC system (Thermo Fisher Scientific). The eluted peptides were detected using a quadrupole Orbitrap Exploris 480 hybrid mass spectrometer (Thermo Fisher Scientific) with normal window DIA. The MS1 scan range was set as a full scan with m/z 490–745 at a mass resolution of 15,000 to set an Auto Gain Control (AGC) target for MS1 as $3×10^6$ and a maximum injection time of 23 ms. The MS2 was collected at more than m/z 200 at 30,000 resolutions to set an AGC target of $3×10^6$, maximum injection time of 'auto', and fixed normalized collision energy of 28%. The isolation width for MS2 was set to 4 m/z, and for the 500–740 m/z window pattern, an optimized window arrangement was used in Scaffold DIA (Proteome Software, Inc, Portland, OR, USA). The raw data were searched against an in silico predicted spectral library using DIA-NN (*Demichev et al., 2020*; *Demichev and Chen, 2022*; version:1.8.1, https://github.com/vdemichev/DiaNN). First, an in silico predicted spectral library was generated from the human protein sequence database (UniProt id UP000005640, reviewed, canonical, 20,381 entries) using DIA-NN. The DIA-NN search parameters were as follows: protease, trypsin; missed cleavages, 1; peptide length range, 7–45; precursor charge range, 2–4; precursor mass range, 495–745; fragment ion m/z range, 200–1800; mass accuracy, 10 ppm; static modification, cysteine carbamidomethylation; enabled 'Heuristic protein interferences', 'Use isotopologues', 'MBR', and 'No shared spectra'. Additional commands were set as follows: 'mass acc cal 10', 'peak translation', and 'matrix spec q'. The protein identification threshold was set at <1% for both peptide and protein false discovery rates (FDRs).

The signal intensity data for 4241 proteins (*Source data 2*) was used for further data analysis below.

## Data analysis

The analysis and visualization of RNA-seq data and proteome data were conducted using custom Python code, and coding and execution were carried out in the Collaboratory. In the RNAseq data, 6685 protein-coding genes were detected (*Source data 1*). Among these, 4291 genes assigned to various categories of KEGG Orthology level 3 (*Kanehisa et al., 2023*) were placed into 77 categories. Analysis was performed on 47 categories that contained more than 10 gene elements. We investigated whether the expression of genes in each category, as a group, was significantly higher or lower upon overexpression of mox and mox-YG compared to the vector control. Specifically, mox/Vector (log2FC) and mox-YG/Vector (log2FC) were calculated along with their FDR values, and each gene was assigned to a KEGG orthology level 3 category. Next, whether the gene groups within each category showed significantly higher or lower expression changes compared to gene groups outside of their category was tested using the Mann-Whitney *U*-test. Significant differences were indicated in the graphs with a red border and red stripes. The graphs showed the overall distribution of

genes within each category using violin plots, and only genes showing significant differences in mox/Vector (log2FC) and mox-YG/Vector (log2FC) were highlighted in swarm plots. An asterisk indicated significant differences on the graphs when the expression changes (log2FC) between mox/Vector and mox-YG/Vector were significant by the Mann-Whitney *U*-test. The significance threshold was set at FDR <0.05 for all. Only categories where either mox or mox-YG overexpression showed significant expression changes were extracted and presented as *Figure 3B*, and all 47 categories were shown in *Figure 3—figure supplement 1C*.

The proteome analysis was conducted using the signal intensity data (*Source data 2*). This data includes proteins that were detected only in some samples under certain conditions or replicates, especially those with low signals or large expression variations. Additionally, the total protein signal sum excluding mox-YG was almost the same between the vector control and mox-YG overexpression (*Figure 4—figure supplement 3A*). This result contradicts the SDS–PAGE findings, which indicate that the total protein mass per cell, including mox-YG, is the same between the vector control and mox-YG overexpression (*Figure 4A*). Therefore, it is considered necessary to adjust the total protein signal sum, including mox-YG, to match that of the vector control to understand the proteome change per cell. In this study, however, we used the most conservative approach by removing proteins with zero values and not applying any overall corrections (*Figure 4—figure supplement 3B*), which allowed the analysis of 3588 proteins. Although this may not correctly analyze 'per cell' proteome change, it enables us to capture the change in proteome composition (excluding mox-YG), thus providing information about cellular responses. If the total correction is applied, only 138 proteins were increased (*Figure 4—figure supplement 3C*), most of which are mitochondrial proteins.

For the detected 3588 proteins above, we compared the increase or decrease of protein groups within each category of KEGG Orthology Level 3 with those outside the category. Analysis was performed on 45 categories, each containing more than 10 proteins, to investigate whether the expression of protein groups in each category was significantly higher or lower under mox-YG overexpression compared to the vector control. Specifically, the average log2FC and its FDR values were calculated using data from three replicates for mox-YG/Vector, and each protein was assigned to a KEGG orthology level 3 category. Next, whether the protein groups within each category showed significantly higher or lower expression changes compared to protein groups outside of their category was tested using the Mann-Whitney *U*-test. Significant differences were indicated in the graphs with a red border and red stripes. The graphs showed the overall distribution of proteins within each category using violin plots, and only proteins showing significant differences in mox-YG/Vector (log2FC) were highlighted in swarm plots. The significance threshold was set at FDR <0.05 for all. Only categories where mox-YG overexpression showed significant expression changes were extracted and presented in *Figure 4C* and all 47 categories were shown in *Figure 4—figure supplement 6*. Published data (*Gowans et al., 2018*) was used to separate protein groups into rapamycin-responsive and non-responsive, and similar analysis and visualization were conducted (*Figure 4—figure supplement 6A, B*).

## Fluorescence microscopic observation

Cells overexpressing target proteins were cultured in SC–LeuUra medium and observed under an inverted microscope (DMI 6000 B, Leica) at log phase ($OD_{660}$=1.0). Images were acquired with Leica Application Suite and processed with Leica Application Suite X software (Leica Microsystems). Cells in brightfield images were separated from the images using yeast_segmentation-master_v3 (*Lu et al., 2019*). Cell morphological traits were assessed using parameters obtained when analyzed using CellProfiler (Ver. 4.2.6) (*Carpenter et al., 2006*). 'AreaShape_Area' was used to analyze FP-overexpressing cell size (arbitrary unit; AU) in *Figure 2—figure supplement 6B*, Nsr1-GFP in *Figure 5—figure supplement 2C* and Nsr1-mScarlet-I in *Figure 5—figure supplements 5E and 6F*, and *Figure 6—figure supplement 7C*. For analysis of the fluorescence intensity of Rip1-mScarlet-I in *Figure 4—figure supplement 8B*, *Figure 6—figure supplement 5A*. 'Intensity_ IntegratedIntensity' was used. 'Intensity_MeanIntensity' was used to analyze the fluorescence intensity of Nsr1-GFP in *Figure 5—figure supplement 2D*, *Figure 6—figure supplement 7C*. Ten images per experiment were used for analysis. The axis ratio in *Figure 6—figure supplement 3* was calculated as the ratio of 'AreaShape_MajorAxisLength' to 'AreaShape_MinorAxisLength' in the data.

## Measurement of oxygen consumption

Cells overexpressing the target protein were collected at $OD_{660}=1$. The cells (0.05 OD unit) were treated with oxygen probe in SC–LeuUra and 50 µM antimycin A in SC–LeuUra. Oxygen consumption was measured according to the instruction manual of the Extracellular OCR Plate Assay Kit (Dojindo, #E297). The fluorescence intensity (Ex500nm/Em650 nm) was recorded at 10 min intervals using a microplate reader (Infinite F200, TECAN). The oxygen consumption rate (AU) was calculated as the slope of a linear approximation of the kinetic data.

## Electron microscopic observation

Cells grown to log phase ($OD_{600}=1.0$) at 30 °C in SC–LeuUra medium were collected and transported to Tokai Electron Microscope Co. Transported samples were sandwiched between copper plates, flash-frozen with liquefied propane, and then dehydrated with anhydrous ethanol. Cells were fixed with a 5:5 mixture of propylene oxide and resin (Quetol-651, Nissin EM Co.). Fixed cells were cut with an ultramicrotome (Ultracut UCT, Leica) to prepare 80-nm-thick sections, stained with 2% uranyl acetate and lead stain solution (Sigma-Aldrich Co.), and observed with a transmission electron microscope (JEM-1400 Plus, JEOL Ltd.).

## Material availability

The strains and plasmids generated in this study are available from NBRP-yeast (https://yeast.nig.ac.jp/yeast/).

## Acknowledgements

We thank the members of the Moriya laboratory (Okayama University) for their helpful discussions. This work was partly supported by the Nagase Science and Technology Foundation, Institute for Fermentation, Osaka (IFO) General Research Funding, JSPS KAKENHI Grant Numbers 22K19294, and 24K02313, JST SPRING Japan Grant Number JPMJSP2126.

## Additional information

### Funding

| Funder | Grant reference number | Author |
| --- | --- | --- |
| Nagase Science Technology Foundation | | Hisao Moriya |
| Institute for Fermentation, Osaka | G-2024-3-022 | Hisao Moriya |
| Japan Society for the Promotion of Science | 22K19294 | Hisao Moriya |
| Japan Society for the Promotion of Science | 24K02313 | Hisao Moriya |
| Japan Science and Technology Agency | JPMJSP2126 | Yuri Fujita |

The funders had no role in study design, data collection and interpretation, or the decision to submit the work for publication.

### Author contributions

Yuri Fujita, Investigation, Visualization, Writing - original draft, Writing – review and editing; Shotaro Namba, Data curation, Writing – review and editing; Yoshiaki Kamada, Investigation; Hisao Moriya, Conceptualization, Supervision, Visualization, Writing - original draft, Writing – review and editing

### Author ORCIDs

Yuri Fujita http://orcid.org/0009-0007-3409-1739
Shotaro Namba https://orcid.org/0000-0002-3516-2347

Yoshiaki Kamada ![ORCID] https://orcid.org/0000-0001-7395-660X
Hisao Moriya ![ORCID] https://orcid.org/0000-0001-7638-3640

Reviewer #1 (Public Review): https://doi.org/10.7554/eLife.99572.3.sa1
Reviewer #2 (Public Review): https://doi.org/10.7554/eLife.99572.3.sa2
Reviewer #3 (Public Review): https://doi.org/10.7554/eLife.99572.3.sa3
Author response https://doi.org/10.7554/eLife.99572.3.sa4

## Additional files

### Supplementary files
MDAR checklist

Source data 1. Processed RNA-seq data.

Source data 2. Raw intensity data of proteome analysis.

### Data availability
Sequencing data have been deposited in DDBJ (accession number: PRJDB18064). All raw data, analysis code, and omics analysis pipelines used for generating the figures have been deposited in GitHub (copy archived at *Moriya, 2025*).

The following dataset was generated:

| Author(s) | Year | Dataset title | Dataset URL | Database and Identifier |
|---|---|---|---|---|
| Moriya H | 2025 | Analysis of the effects of heterologous protein expression on budding yeast cells | https://ddbj.nig.ac.jp/search/entry/bioproject/PRJDB18064 | DNA Data Bank of Japan, PRJDB18064 |

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

# Appendix 1

## Appendix 1—key resources table

| Reagent type (species) or resource | Designation | Source or reference | Identifiers | Additional information |
|---|---|---|---|---|
| Strain, strain background (*Saccharomyces cerevisiae*) | BY4741 MATa his3Δ1 leu2Δ0 met15Δ0 ura3Δ0 | PMID:9483801 | | |
| Strain (ACT1pro_mScarlet-I) | fcy1::ACT1pro_mScarlet-I_TDH3ter_HIS3MX6 | this paper | | |
| Strain (TDH3pro_mScarlet-I) | fcy1::TDH3pro_mScarlet-I_TDH3ter_HIS3MX6 | this paper | | |
| Strain (YAP5pro_mScarlet-I) | fcy1::YAP5pro_mScarlet-I_TDH3ter_HIS3MX6 | this paper | | |
| Strain (PHO84pro_mScarlet-I) | fcy1::PHO84pro_mScarlet-I_TDH3ter_HIS3MX6 | this paper | | |
| Strain (ZPS1pro_mScarlet-I) | fcy1::ZPS1pro_mScarlet-I_TDH3ter_HIS3MX6 | this paper | | |
| Strain (ADE17pro_mScarlet-I) | fcy1::ADE17pro_mScarlet-I_TDH3ter_HIS3MX6 | this paper | | |
| Strain (CTR1pro_mScarlet-I) | fcy1::CTR1pro_mScarlet-I_TDH3ter_HIS3MX6 | this paper | | |
| Strain (FIT2pro_mScarlet-I) | fcy1::FIT2pro_mScarlet-I_TDH3ter_HIS3MX6 | this paper | | |
| Strain (HXT7pro_mScarlet-I) | fcy1::HXT7pro_mScarlet-I_TDH3ter_HIS3MX6 | this paper | | |
| Strain (CUP1pro_mScarlet-I) | fcy1::CUP1pro_mScarlet-I_TDH3ter_HIS3MX6 | this paper | | |
| Strain (OM14pro_mScarlet-I) | fcy1::OM14pro_mScarlet-I_TDH3ter_HIS3MX6 | this paper | | |
| Strain (OM45pro_mScarlet-I) | fcy1::OM45pro_mScarlet-I_TDH3ter_HIS3MX6 | this paper | | |
| Strain (AGP1pro_mScarlet-I) | fcy1::AGP1pro_mScarlet-I_TDH3ter_HIS3MX6 | this paper | | |
| Strain (GAP1pro_mScarlet-I) | fcy1::GAP1pro_mScarlet-I_TDH3ter_HIS3MX6 | this paper | | |
| Strain (HXT1pro_mScarlet-I) | fcy1::HXT1pro_mScarlet-I_TDH3ter_HIS3MX6 | this paper | | |
| Strain (RIP1pro_RIP1_mScarlet-I) | fcy1::RIP1pro_RIP1_mScarlet-I_TDH3ter_HIS3MX6 | this paper | | |
| Strain (NSR1pro_NSR1_GFP) | fcy1::NSR1pro_NSR1_GFP_TDH3ter_HIS3MX6 | this paper | | |
| Strain (NSR1pro_NSR1_mScarlet-I) | fcy1::NSR1pro_NSR1_mScarlet-I_TDH3ter_HIS3MX6 | this paper | | |
| Strain (ACT1pro_GBP_mScarlet-I) | fcy1::ACT1pro_GBP_mScarlet-I_TDH3ter_HIS3MX6 | this paper | | |
| Strain, strain background (*Saccharomyces cerevisiae*) | mtr4-1 MATa his3mtr4-1::KanR; his3Δ1 leu2Δ0 ura3Δ0 met15Δ0 | PMID:27708008 | | |
| Strain (mtr4 ts mut, NSR1pro_NSR1_mScarlet-I) | mtr4-1 MATa his3mtr4-1::KanR; his3Δ1 leu2Δ0 ura3Δ0 met15Δ0 fcy1::NSR1pro_NSR1_mScarlet-I_TDH3ter_HIS3MX6 | this paper | | |
| Gene (*S. cerevisiae*) | GPM1 | PMID:30095406 | SGD:YKL152C | |
| Gene (*S. cerevisiae*) | TDH3 | PMID:30095406 | SGD: YGR192C | |
| Genetic reagent (*S. cerevisiae*) | GPM1-H182A | PMID:1386023 | | catalytic center mutant of GPM1 |
| Genetic reagent (*S. cerevisiae*) | TDH3-C150S | PMID:25580853 | | catalytic center mutant of TDH3 |
| Genetic reagent (*S. cerevisiae*) | EGFP | PMID:9043107 | | |
| Genetic reagent (*S. cerevisiae*) | EGFP-Y66G | PMID:27538565 | | non-fluorescent yEGFP mutant |

*Appendix 1 Continued on next page*

*Appendix 1 Continued*

| Reagent type (species) or resource | Designation | Source or reference | Identifiers | Additional information |
|---|---|---|---|---|
| Genetic reagent (*S. cerevisiae*) | sfGFP | PMID:16369541 | | |
| Genetic reagent (*S. cerevisiae*) | sfGFP-Y66G | this paper | | non-fluorescent sfGFP mutant |
| Genetic reagent (*S. cerevisiae*) | moxGFP | PMID:26158227 | | |
| Genetic reagent (*S. cerevisiae*) | moxGFP-Y66G | this paper | | non-fluorescent moxGFP mutant |
| Genetic reagent (*S. cerevisiae*) | moxGFP-Y66G+Y | this paper | | Mutant with Y added to the end of the moxGFP-Y66G sequence |
| Genetic reagent (*S. cerevisiae*) | moxGFP-T203I | this paper | | Mutant with weaker fluorescence than moxGFP |
| Genetic reagent (*S. cerevisiae*) | moxGFP-T65S | this paper | | Mutant with weaker fluorescence than moxGFP |
| Genetic reagent (*S. cerevisiae*) | mCherry | PMID:15558047 | | |
| Genetic reagent (*S. cerevisiae*) | mCherry-Y72G | this paper | | non-fluorescent mCherry mutant |
| Genetic reagent (*S. cerevisiae*) | mCherry-Kafri | PMID:26725116 | | |
| Genetic reagent (*S. cerevisiae*) | mCherry-Kafri-Y72G | this paper | | non-fluorescent mCherry-Kafri mutant |
| Genetic reagent (*S. cerevisiae*) | mScarlet-I | PMID:27869816 | | |
| Genetic reagent (*S. cerevisiae*) | GFP binding protein (GBP) | PMID:17060912 | | |
| Gene (*S. cerevisiae*) | ACT1 | | SGD: YFL039C | |
| Gene (*S. cerevisiae*) | YAP5 | | SGD: YIR018W | |
| Gene (*S. cerevisiae*) | PHO84 | | SGD: YML123C | |
| Gene (*S. cerevisiae*) | ZPS1 | | SGD: YOL154W | |
| Gene (*S. cerevisiae*) | ADE17 | | SGD: YMR120C | |
| Gene (*S. cerevisiae*) | CTR1 | | SGD: YPR124W | |
| Gene (*S. cerevisiae*) | FIT2 | | SGD: YOR382W | |
| Gene (*S. cerevisiae*) | HXT7 | | SGD: YDR342C | |
| Gene (*S. cerevisiae*) | CUP1 | | SGD: YHR094C | |
| Gene (*S. cerevisiae*) | OM14 | | SGD: YBR230C | |
| Gene (*S. cerevisiae*) | OM45 | | SGD: YIL136W | |
| Gene (*S. cerevisiae*) | AGP1 | | SGD: YCL025C | |
| Gene (*S. cerevisiae*) | GAP1 | | SGD: YKR039W | |
| Gene (*S. cerevisiae*) | HXT1 | | SGD: YHR094C | |
| Gene (*S. cerevisiae*) | RIP1 | | SGD: YEL024W | |
| Gene (*S. cerevisiae*) | NSR1 | | SGD: YGR159C | |
| Gene (*S. cerevisiae*) | ATG13 | | SGD: YPR185W | |
| Recombinant DNA reagent | pTOW40836 | PMID:22722869 | | 2µOri, URA3, leu2d, AmpR, ColE1Ori |
| Recombinant DNA reagent | pTOW-t-EGFP | PMID:27538565 | | TDH3 promoter yEGFP; background pTOW40836 |
| Recombinant DNA reagent | pTOW-t-EGFP-Y66G | PMID:27538565 | | TDH3 promoter yEGFP-Y66G; background pTOW40836 |
| Recombinant DNA reagent | pTOW-t-sfGFP | PMID:30095406 | | TDH3 promoter sfGFP; background pTOW40836 |
| Recombinant DNA reagent | pTOW-t-sfGFP-Y66G | this paper | | TDH3 promoter sfGFP-Y66G; background pTOW40836 |
| Recombinant DNA reagent | pTOW-t-mox | PMID:30095406 | | TDH3 promoter moxGFP; background pTOW40836 |
| Recombinant DNA reagent | pTOW-t-mox-Y66G | this paper | | TDH3 promoter moxGFP-Y66G; background pTOW40836 |
| Recombinant DNA reagent | pTOW-t-moxFS | this paper | | TDH3 promoter moxFS; background pTOW40836 |
| Recombinant DNA reagent | pTOW-t-moxGFP-T203I | this paper | | TDH3 promoter moxGFP-T203I; background pTOW40836 |
| Recombinant DNA reagent | pTOW-t-moxGFP-T65S | this paper | | TDH3 promoter moxGFP-T65S; background pTOW40836 |
| Recombinant DNA reagent | pTOW-t-mCherry | this paper | | TDH3 promoter mCherry; background pTOW40836 |

Appendix 1 Continued

| Reagent type (species) or resource | Designation | Source or reference | Identifiers | Additional information |
|---|---|---|---|---|
| Recombinant DNA reagent | pTOW-t-mCherry-Y72G | this paper | | TDH3 promoter mCherry-Y72G; background pTOW40836 |
| Recombinant DNA reagent | pTOW-t-mCherry-Kafri | this paper | | TDH3 promoter mCherry-Kafri; background pTOW40836 |
| Recombinant DNA reagent | pTOW-t-mCherry-Kafri-Y72G | this paper | | TDH3 promoter mCherry-Kafri-Y72G; background pTOW40836 |
| Recombinant DNA reagent | pTOW-t-GPM1 | PMID:30095406 | | TDH3 promoter GPM1; background pTOW40836 |
| Recombinant DNA reagent | pTOW-t-TDH3 | PMID:30095406 | | TDH3 promoter TDH3; background pTOW40836 |
| Recombinant DNA reagent | pTOW-t-GPM1-H182A | PMID:30095406 | | TDH3 promoter GPM1-H182A; background pTOW40836 |
| Recombinant DNA reagent | pTOW-t-TDH3-C150S | PMID:30095406 | | TDH3 promoter TDH3-C150S; background pTOW40836 |
| Recombinant DNA reagent | pTOW4083-AUR1d | this paper | | 2μOri, URA3, aur1d, AmpR, ColE1Ori |
| Recombinant DNA reagent | pTOW4083-AUR1d-mox | this paper | | TDH3 promoter moxGFP; background pTOW40836-AUR1d |
| Recombinant DNA reagent | pTOW4083-AUR1d-moxY66G | this paper | | TDH3 promoter moxGFP-Y66G; background pTOW40836-AUR1d |
| Recombinant DNA reagent | pTOW4083-AUR1d-mCherry | this paper | | TDH3 promoter mCherry; background pTOW40836-AUR1d |
| Recombinant DNA reagent | pTOW4083-AUR1mCherry-Y72G | this paper | | TDH3 promoter mCherry-Y72G; background pTOW40836-AUR1d |
| Recombinant DNA reagent | pRS423ks | PMID:23275495 | | 2μOri, HIS3, AmpR, ColE1Ori |
| Recombinant DNA reagent | pRS423ks-ATG13 | this paper | | ATG13promoter ATG13; background pRS423ks |
| Sequence-based reagent | AUR1d | this paper | | (sequence) |
| Sequence-based reagent | EGFP | PMID:9043107 | | (sequence) |

Appendix 1 Continued on next page

*Appendix 1 Continued*

| Reagent type (species) or resource | Designation | Source or reference | Identifiers | Additional information |
|---|---|---|---|---|
| Sequence-based reagent | EGFP-Y66G | this paper | | ATGTCTAAAGGTGAAGAATTATTCACTGGTGTTGTCCCAATTTTGGTTGAATTAGATGGTGA TGTTAATGGTCACAAATTTTCTGTCTCCGGTGAAGGTGAAGGTGATGCTACTTACGGTAAA TTGACCTTAAAATTTATTTGTACTACTGGTAAATTGCCAGTTCCATGGCCAACCTTAGTC ACTACTTTAACTggTGGTGTTCAATGTTTTTCTAGATACCCAGATCATATGAAACAACATGA CTTTTTCAAGTCTGCCATGCCAGAAGGTTATGTTCAAGAAAGAACTTTTTTTCAAAGATG ACGGTAACTACAAGACCAGAGCTGAAGTCAAGTTTGAAGGTGATACCTTAGTTAATAG AATCGAATTAAAAGGTATTGATTTTAAAGAAGATGGTAACATTTTAGGTCACAAATTGGAAT ACAACTATAACTCTCACAATGTTTACATCATGGCTGACAAACAAAAGAATGGTATCAAAGTT AACTTCAAAATTAGACACAACATTGAAGATGGTTCTGTTCAATTAGCTGACCATTATCAAC AAAATACTCCAATTGGTGATGGTCCAGTCTTGTTACCAGACAACCATTACTTATCCACT CAATCTGCCTTATCCAAAGATCCAAACGAAAAGAGAGACCACATGGTCTTGTTAGAATT TGTTACTGCTGCTGGTATTACCCATGGTATGGATGAATTGTACAAATAA |
| Sequence-based reagent | sfGFP | PMID:16369541 | | ATGTCCAAGGGTGAAGAGCTATTTACTGGGGTTGTACCCATTTTGGTAGAACTGGACGGA GATGTAAACGGACATAAATTCTCTGTTAGAGGTGAGGGCGAAGGCGATGCCACCAATGGT AAATTGACTCTGAAGTTTATATGCACTACGGGTAAATTACCTGTTCCTTGGCCAACCCTA GTAACAACTTTGACATATGGTGTTCAATGTTTCTCAAGATACCCAGACCATATGAAAAGG CATGATTTCTTTAAAAGTGCTATGCCAGAAGGCTACGTGCAAGAGAGAACTATCTCCTTT AAGGATGACGGTACGTATAAAACACGAGCAGAAGTGAAATTCGAAGGGGGATACACTAGTT AATCGCATCGAATTAAAGGGTATAGACTTTAAGGAAGATGGTAATATTCTCGGCCATAAA CTTGAGTATAATTTCAACTCGCATAATGTGTACATTACAGCTGACAAACAAAAGAACGGA ATTAAAGCGAATTTTAAAATCAGGCACAACGTCGAAGATGGGTCTGTTCAACTTGCCGAT CATTATCAGCAAAACACCCCTATTGGTGATGGTCCAGTCTTGTTACCCGATAATCACTAC TTAAGCACACAGTCTAGATTGTCAAAAGATCCGAATGAAAAGCGTGATCACATGGTTTTA TTGGAATTTGTCACCGCTGCAGGAATAACTCACGGAATGGACGAGCTTTATAAGTAA |
| Sequence-based reagent | sfGFP-Y66G | this paper | | ATGTCCAAGGGTGAAGAGCTATTTACTGGGGTTGTACCCATTTTGGTAGAACTGGAC GGAGATGTAAACGGACATAAATTCTCTGTTAGAGGTGAGGGCGAAGGCGATGCCACCA ATGGTAAATTGACTCTGAAGTTTATATGCACTACGGGTAAATTACCTGTTCCTTGGCCAAC CCTAGTAACAACTTTGACAggTGGTGTTCAATGTTTCTCAAGATACCCAGACCATATGAA AAGGCATGATTTCTTTAAAAGTGCTATGCCAGAAGGCTACGTGCAAGAGAGAACTATCT CCTTTAAGGATGACGGTACGTATAAAACACGAGCAGAAGTGAAATTCGAAGGGGGATACA CTAGTTAATCGCATCGAATTAAAGGGTATAGACTTTAAGGAAGATGGTAATATTCTCGGCCA TAAACTTGAGTATAATTTCAACTCGCATAATGTGTACATTACAGCTGACAAACAAAAGAAC GGAATTAAAGCGAATTTTAAAATCAGGCACAACGTCGAAGATGGGTCTGTTCAACTTGC CGATCATTATCAGCAAAACACCCCTATTGGTGATGGTCCAGTCTTGTTACCCGATAATCA CTACTTAAGCACACAGTCTAGATTGTCAAAAGATCCGAATGAAAAGCGTGATCACATGGTT TTATTGGAATTTGTCACCGCTGCAGGAATAACTCACGGAATGGACGAGCTTTATAAGTAA |
| Sequence-based reagent | moxGFP | PMID:30095406 | | ATGTCTAAAGGTGAAGAATTATTCACTGGTGTTGTCCCAATTTTGGTTGAATTAGATGGTGA TGTTAATGGTCACAAATTTTCTGTCcgtGGTGAAGGTGAAGGTGATGCTACTaAtGGTAA ATTGACCTTAAAATTTATTTcTACTACTGGTAAATTGCCAGTTCCATGGCCAACCTTAGTCACT ACTTTAACTTATGGTGTTCAATcTTTTTCTAGATACCCAGATCATATGAAACgtCATGACTT TTTCAAGTCTGCCATGCCAGAAGGTTATGTTCAAGAAAGAACTATTTcTTTCAAAGATGACG GTAActTACAAGACCAGAGCTGAAGTCAAGTTTGAAGGTGATACCTTAGTTAATAGAATC GAATTAAAAGGTATTGATTTTAAAGAAGATGGTAACATTTTAGGTCACAAATTGGAATACAA CTtcAACTCTCACAATGTTTACATCActGCTGACAAACAAAAGAATGGTATCAAAGcTAA CTTCAAAATTAGACACAACgtTGAAGATGGTTCTGTTCAATTAGCTGACCATTATCAACA AAATACTCCAATTGGTGATGGTCCAGTCTTGTTACCAGACAACCATTACTTATCCACTC AATCTcgtTTATCCAAAGATCCAAACGAAAAGAGAGACCACATGGTCTTGTTAGAATTTG TTACTGCTGCTGGTATTACCCATGGTATGGATGAATTGTACAAATAA |
| Sequence-based reagent | mox-Y66G | this paper | | ATGTCTAAAGGTGAAGAATTATTCACTGGTGTTGTCCCAATTTTGGTTGAATTAGATGG TGATGTTAATGGTCACAAATTTTCTGTCcgtGGTGAAGGTGAAGGTGATGCTACTaAtGGTAA ATTGACCTTAAAATTTATTTcTACTACTGGTAAATTGCCAGTTCCATGGCCAACCTTAGTC ACTACTTTAACTggTGGTGTTCAATcTTTTTCTAGATACCCAGATCATATGAAACgtCATG ACTTTTTCAAGTCTGCCATGCCAGAAGGTTATGTTCAAGAAAGAACTATTTcTTTCAAAG ATGACGGTActTACAAGACCAGAGCTGAAGTCAAGTTTGAAGGTGATACCTTAGTTAAT AGAATCGAATTAAAAGGTATTGATTTTAAAGAAGATGGTAACATTTTAGGTCACAAATTG GAATACAACTtcAACTCTCACAATGTTTACATCActGCTGACAAACAAAAGAATGGTATC AAAGcTAACTTCAAAATTAGACACAACgtTGAAGATGGTTCTGTTCAATTAGCTGACCAT TATCAACAAAATACTCCAATTGGTGATGGTCCAGTCTTGTTACCAGACAACCATTACTTA TCCACTCAATCTcgtTTATCCAAAGATCCAAACGAAAAGAGAGACCACATGGTCTTGTTA GAATTTGTTACTGCTGCTGGTATTACCCATGGTATGGATGAATTGTACAAATAA |
| Sequence-based reagent (synthetic gene) | mCherry(yeast codon-optimized) | this paper | | ATGGTTTCTAAAGGTGAAGAAGATAATATGGCTATTATTAAAGAATTTATGAGATTTAAAG TTCATATGGAAGGTTCAGTTAATGGTCATGAATTTGAAATTGAAGGTGAAGGTGAAGG TAGACCATATGAAGGTACTCAAACTGCTAAATTGAAAGTTACTAAAGGTGGTCCATTAC CATTTGCTTGGGATATTTTGTCACCACAATTTATGTATGGTTCAAAAGCTTATGTTAAACAT CCAGCTGATATTCCAGATTATTTAAAATTGTCATTTCCAGAAGGTTTTAAATGGGAAAGAG TTATGAATTTTGAAGATGGTGGTGTTGTTACTGTTACTCAAGATTCATCATTACAAGATGGT GAATTTATTTATAAAGTTAAATTGAGAGGTACTAATTTTCCATCAGATGGTCCAGTTATGC AAAAAAAAAACTATGGGTTGGGAAGCTTCATCAGAAAGAATGTATCCAGAAGATGGTGC TTTAAAAGGTGAAATTAAACAAAGATTGAAATTAAAAGATGGTGGTCATTATGATGCTG AAGTTAAAACTACTTATAAAGCTAAAAAACCAGTTCAATTACCAGGTGCTTATAATGTTAA TATTAAATTGGATATTACTTCACATAATGAAGATTATACTATTGTTGAACAATATGAAAGAG CTGAAGGTAGACATTCAACTGGTGGTATGGATGAATTGTACAAATAA |

*Appendix 1 Continued on next page*

*Appendix 1 Continued*

| Reagent type (species) or resource | Designation | Source or reference | Identifiers | Additional information |
|---|---|---|---|---|
| Sequence-based reagent | mCherry-Y72G | this paper | | gaataaacacacataaacaaacaaaATGGTGAGCAAGGGCGAGGAGGATAACATGGCCATCATCAAGGAGTTCATGCGCTTCAAGGTGCACATGGAGGGCTCCGTGAACGGCCACGAGTTCGAGATCGAGGGCGAGGGCGAGGGCCGCCCCTACGAGGGCACCCAGACCGCCAAGCTGAAGGTGACCAAGGGTGGCCCCCTGCCCCTTCGCCTGGGACATCCTGTCCCCTCAGTTCATGggCGGCTCCAAGGCCTACGTGAAGCACCCCGCCGACATCCCCGACTACTTGAAGCTGTCCTTCCCCGAGGGCTTCAAGTGGGAGCGCGTGATGAACTTCGAGGACGGCGGCGTGGTGACCGTGACCCAGGACTCCTCCCTGCAGGACGGCGAGTTCATCTACAAGGTGAAGCTGCGCGGCACCAACTTCCCCCTCCGACGGCCCCGTAATGCAGAAGAAGACCATGGGCTGGGAGGCCTCCTCCGAGCGGATGTACCCCGAGGACGGCGCCCTGAAGGGCGAGATCAAGCAGAGGCTGAAGCTGAAGGACGGCGGCCACTACGACGCTGAGGTCAAGACCACCTACAAGGCCAAGAAGCCCGTGCAGCTGCCCGGCGCCTACAACGTCAACATCAAGTTGGACATCACCTCCCACAACGAGGACTACACCATCGTGGAACAGTACGAACGCGCCGAGGGCCGCCACTCCACCGGCGGCATGGACGAGCTGTACAAGTAGgtgaatttactttaaatcttgcatt |
| Sequence-based reagent (synthetic gene) | mCherry-Kafri | PMID:26725116 | | ATGGTGAGCAAGGGCGAGGAGGATAACATGGCCATCATCAAGGAGTTCATGCGCTTCAAGGTGCACATGGAGGGCTCCGTGAACGGCCACGAGTTCGAGATCGAGGGCGAGGGCGAGGGCCGCCCCTACGAGGGCACCCAGACCGCCAAGCTGAAGGTGACCAAGGGTGGCCCCCTGCCCCTTCGCCTGGGACATCCTGTCCCCTCAGTTCATGTACGGCTCCAAGGCCTACGTGAAGCACCCCGCCGACATCCCCGACTACTTGAAGCTGTCCTTCCCCGAGGGCTTCAAGTGGGAGCGCGTGATGAACTTCGAGGACGGCGGCGTGGTGACCGTGACCCAGGACTCCTCCCTGCAGGACGGCGAGTTCATCTACAAGGTGAAGCTGCGCGGCACCAACTTCCCCCTCCGACGGCCCCGTAATGCAGAAGAAGACCATGGGCTGGGAGGCCTCCTCCGAGCGGATGTACCCCGAGGACGGCGCCCTGAAGGGCGAGATCAAGCAGAGGCTGAAGCTGAAGGACGGCGGCCACTACGACGCTGAGGTCAAGACCACCTACAAGGCCAAGAAGCCCGTGCAGCTGCCCGGCGCCTACAACGTCAACATCAAGTTGGACATCACCTCCCACAACGAGGACTACACCATCGTGGAACAGTACGAACGCGCCGAGGGCCGCCACTCCACCGGCGGCATGGACGAGCTGTACAAGTAG |
| Sequence-based reagent | mCherry-Kafri-Y72G | this paper | | ATGGTGAGCAAGGGCGAGGAGGATAACATGGCCATCATCAAGGAGTTCATGCGCTTCAAGGTGCACATGGAGGGCTCCGTGAACGGCCACGAGTTCGAGATCGAGGGCGAGGGCGAGGGCCGCCCCTACGAGGGCACCCAGACCGCCAAGCTGAAGGTGACCAAGGGTGGCCCCCTGCCCCTTCGCCTGGGACATCCTGTCCCCTCAGTTCATGggCGGCTCCAAGGCCTACGTGAAGCACCCCGCCGACATCCCCGACTACTTGAAGCTGTCCTTCCCCGAGGGCTTCAAGTGGGAGCGCGTGATGAACTTCGAGGACGGCGGCGTGGTGACCGTGACCCAGGACTCCTCCCTGCAGGACGGCGCCCTGAAGGGCGAGATCAAGCAGAGGCTGAATGCAGAAGAAGACCATGGGCTGGGAGGCCTCCTCCGAGCGGATGTACCCCGAGGACGGCGCCCTGAAGGGCGAGATCAAGCAGAGGCTGAAGCTGAAGGACGGCGGCCACTACGACGCTGAGGTCAAGACCACCTACAAGGCCAAGAAGCCCGTGCAGCTGCCCGGCGCCTACAACGTCAACATCAAGTTGGACATCACCTCCCACAACGAGGACTACACCATCGTGGAACAGTACGAACGCGCCGAGGGCCGCCACTCCACCGGCGGCATGGACGAGCTGTACAAGTAG |
| Sequence-based reagent | moxFS | this paper | | ATGcgcaTCTAAAGGTGAAGAATTATTCACTGGTGTTGTCCCAATTTTGGTTGAATTAGATGGTGATGTTAATGGTCACAAATTTTCTGTCcgtGGTGAAGGTGAAGGTGATGCTACTaAtGGTAAATTGACCTTAAAATTTATTtCTACTACTGGTAAATTGCCAGTTCCATGGCCAACCTTAGTCACTACTTTAACTTATGGTGTTCAATcTTTTTCTAGATACCCAGATCATATGAAACgtCATGACTTTTTCAAGTCTGCCATGCCAGAAGGTTATGTTCAAGAAAGAACTATTTcTTTCAAAGATGACGGTActACAAGACCAGAGCTGAAGTCAAGTTTGAAGGTGATACCTTAGTTAATAGAATCGAATTAAAAGGTATTGATTTTAAAGAAGATGGTAACATTTTAGGTCACAAATTGGAATACAACTtcAACTCTCACAATGTTTACATCActGCTGACAAACAAAAGAATGGTATCAAAGcTAACTTCAAAATTAGACACAACgtTGAAGATGGTTCTGTTCAATTAGCTGACCATTATCAACAAAATACTCCAATTGGTGATGGTCCAGTCTTGTTACCAGACAACCATTACTTATCCACTCAATCTcgtTTATCCAAAGATCCAAACGAAAAGAGAGACCACATGGTCTTGTTAGAATTTGTTACTGCTGCTGGTATTACCCATGGTATGGATGAATTGTACAAATAA |
| Sequence-based reagent | mox-Y66G+Y | this paper | | ATGTCTAAAGGTGAAGAATTATTCACTGGTGTTGTCCCAATTTTGGTTGAATTAGATGGTGATGTTAATGGTCACAAATTTTCTGTCcgtGGTGAAGGTGAAGGTGATGCTACTaAtGGTAAATTGACCTTAAAATTTATTtCTACTACTGGTAAATTGCCAGTTCCATGGCCAACCTTAGTCACTACTTTAACTggTGGTGTTCAATcTTTTTCTAGATACCCAGATCATATGAAACgtCATGACTTTTTCAAGTCTGCCATGCCAGAAGGTTATGTTCAAGAAAGAACTATTTcTTTCAAAGATGACGGTActACAAGACCAGAGCTGAAGTCAAGTTTGAAGGTGATACCTTAGTTAATAGAATCGAATTAAAAGGTATTGATTTTAAAGAAGATGGTAACATTTTAGGTCACAAATTGGAATACAACTtcAACTCTCACAATGTTTACATCActGCTGACAAACAAAAGAATGGTATCAAAGcTAACTTCAAAATTAGACACAACgtTGAAGATGGTTCTGTTCAATTAGCTGACCATTATCAACAAAATACTCCAATTGGTGATGGTCCAGTCTTGTTACCAGACAACCATTACTTATCCACTCAATCTcgtTTATCCAAAGATCCAAACGAAAAGAGAGACCACATGGTCTTGTTAGAATTTGTTACTGCTGCTGGTATTACCCATGGTATGGATGAATTGTACAAATATTAA |
| Sequence-based reagent | GFP-binding protein | PMID:17060912 | | GCTCAAGTTCAATTGGTTGAATCTGGTGGTGCTTTGGTTCAACCAGGTGGTTCTTTGAGATTGTCTTGTGCTGCTTCTGGTTTCCCAGTTAACAGATACTCTATGAGATGGTACAGACAAGCTCCAGGTAAGGAAAGAGAATGGGTTGCTGGTATGTCTTCTGCTGGTGACAGATCTTCTTACGAAGACTCTGTTAAGGGTAGATTCACTATTTCTAGAGACGACGCTAGAAACACTGTTTACTTGCAAATGAACTCTTTGAAGCCAGAAGACACTGCTGTTTACTACTGTAACGTTAACGTTGGTTTCGAATACTGGGGTCAAGGTACTCAAGTTACTGTTTCTTCTAAGTAA |

*Appendix 1 Continued on next page*

Appendix 1 Continued

| Reagent type (species) or resource | Designation | Source or reference | Identifiers | Additional information |
|---|---|---|---|---|
| Sequence-based reagent | mox-T203I | this paper | | ATGTCTAAAGGTGAAGAATTATTCACTGGTGTTGTCCCAATTTTGGTTGAATTAGATGG TGATGTTAATGGTCACAAATTTTCTGTCcgtGGTGAAGGTGAAGGTGATGCTACTaAtGG TAAATTGACCTTAAAATTTATTTcTACTACTGGTAAATTGCCAGTTCCATGGCCAACCTTAGTC ACTACTTTAACTTATGGTGTTCAATcTTTTTCTAGATACCCAGATCATATGAAACgtCATGACT TTTTCAAGTCTGCCATGCCAGAAGGTTATGTTCAAGAAAGAACTATTTcTTTCAAAGATGAC GGTActTACAAGACCAGAGCTGAAGTCAAGTTTGAAGGTGATACCTTAGTTAATAGAATCG AATTAAAAGGTATTGATTTTAAAGAAGATGGTAACATTTTAGGTCACAAATTGGAATACAACT tcAACTCTCACAATGTTTACATCACtGCTGACAAACAAAAGAATGGTATCAAAGcTAACTT CAAAATTAGACACAACgtTGAAGATGGTTCTGTTCAATTAGCTGACCATTATCAACAAAATA CTCCAATTGGTGAGGTCCAGTCTTGTTACCAGACAACCATTACTTATCCAtTCAATCTcgt TTATCCAAAGATCCAAACGAAAAGAGAGACCCACATGGTCTTGTTAGAATTTGTTACTGC TGCTGGTATTACCCATGGTATGGATGAATTGTACAAATAA |
| Sequence-based reagent | mox-T65S | this paper | | ATGTCTAAAGGTGAAGAATTATTCACTGGTGTTGTCCCAATTTTGGTTGAATTAGATGGTGA TGTTAATGGTCACAAATTTTCTGTCcgtGGTGAAGGTGAAGGTGATGCTACTaAtGGTAAAT TGACCTTAAAATTTATTTcTACTACTGGTAAATTGCCAGTTCCATGGCCAACCTTAGTCACT ACTTTATCTTATGGTGTTCAATcTTTTTCTAGATACCCAGATCATATGAAACgtCATGACTT TTTCAAGTCTGCCATGCCAGAAGGTTATGTTCAAGAAAGAACTATTTcTTTCAAAGATGACG GTActTACAAGACCAGAGCTGAAGTCAAGTTTGAAGGTGATACCTTAGTTAATAGAATCGAA TTAAAAGGTATTGATTTTAAAGAAGATGGTAACATTTTAGGTCACAAATTGGAATACAA CTtcAACTCTCACAATGTTTACATCACtGCTGACAAACAAAAGAATGGTATCAAAGcTAACTT CAAAATTAGACACAACgtTGAAGATGGTTCTGTTCAATTAGCTGACCATTATCAACAAA ATACTCCAATTGGTGATGGTCCAGTCTTGTTACCAGACAACCATTACTTATCCACTCAATCTc gtTTATCCAAAGATCCAAACGAAAAGAGAGACCCACATGGTCTTGTTAGAATTTGTTACTG CTGCTGGTATTACCCATGGTATGGATGAATTGTACAAATAA |
| Sequence-based reagent | mScarlet-I | this paper | | ATGGTTTCTAAGGGTGAAGCTGTTATTAAGGAATTCATGAGATTCAAGGTTCACATGG AAGGTTCTATGAACGGTCACGAATTCGAAATTGAAGGTGAAGGTGAAGGTAGACCATACG AAGGTACTCAAACTGCTAAGTTGAAGGTTACTAAGGGTGGTCCATTGCCATTCTCTTGGG ACATTTTGTCTCCACAATTCATGTACGGTTCTAGAGCTTTCAtTAAGCACCCAGCTGACATT CCAGACTACTACAAGCAATCTTTCCCAGAAGGTTTCAAGTGGGAAAGAGTTATGAACTT CGAAGACGGTGGTGCTGTTACTGTTACTCAAGACACTTCTTTGGAAGACGGTACTTTGAT TTACAAGGTTAAGTTGAGAGGTACTAACTTCCCACCAGACGGTCCAGTTATGCAAAAGA AGACTATGGGTTGGGAAGCTTCTACTGAAAGATTGTACCCAGAAGACGGTGTTTTGA AGGGTGACATTAAGCACGCTTTGAGATTGAAGGACGGTGGTAGATACTTGGCTGACTTC AAGACTACTTACAAGGCTAAGAAGCCAGTTCAAATGCCAGGTGCTTACAACGTTGACAGA AAGTTGGACATTACTTCTCACAACGAAGACTACACTGTTGTTGAACAATACGAAAGAT CTGAAGGTAGACACTCTACTGGTGGTATGGACGAATTGTACAAGTAA |
| Sequence-based reagent | GPM1 | PMID:30095406 | | ATGCCAAAGTTAGTTTTAGTTAGACACGGTCAATCCGAATGGAACGAAAAGAACTTATTCA CCGGTTGGGTTGATGTTAAATTGTCTGCCAAGGGTCAACAAGAAGCCGCTAGAGCCGGT GAATTGTTGAAGGAAAAGAAGGTCTACCCAGACGTCTTGTACACTTCCAAGTTGTCC AGAGCTATCCAAACTGCTAACATTGCTTTGGAAAAGGCTGACAGATTATGGATTCCAGTCA ACAGATCCTGGAGATTGAACGAAAGACATTACGGTGACTTACAAGGTAAGGACAAGGCT GAAACTTTGAAGAAGTTCGGTGAAGAAAAATTCAACACCTACAGAAGATCCTTCGATGTT CCACCTCCCCCAATCGACGCTTCTTCTCCATTCTCTCAAAAGGGTGATGAAAGATACAA GTACGTTGACCCAAATGTCTTGCCAGAAACTGAATCTTTGGCTTTGGTCATTGACAGATT GTTGCCATACTGGCAAGATGTCATTGCCAAGGACTTGTTGAGTGGTAAGACCGTCAT GATCGCCGCTCACGGTAACTCCTTGAGAGGTTTGGTTAAGCACTTGGAAGGTATCTC TGATGCTGACATTGCTAAGTTGAACATCCCAACTGGTATTCCATTGGTCTTCGAATTGG ACGAAAACTTGAAGCCATCTAAGCCATCTTACTACTTGGACCCAGAAGCTGCCGCTGC TGGTGCCGCTGCTGTTGCCAACCAAGGTAAGAAATAA |
| Sequence-based reagent | TDH3 | PMID:30095406 | | ATGGTTAGAGTTGCTATTAACGGTTTCGGTAGAATCGGTAGATTGGTCATGAGAATTGCT TTGTCTAGACCAAACGTCGAAGTTGTTGCTTTGAACGACCCATTCATCACCAACGAC TACGCTGCTTACATGTTCAAGTACGACTCCACTCACGGTAGATACGCTGGTGAAGTTTCC CACGATGACAAGCACATCATTGTCGATGGTAAGAAGATTGCTACTTACCAAGAAAGAGAC CCAGCTAACTTGCCATGGGGTTCTTCCAACGTTGACATCGCCATTGACTCCACTGGTGTT TTCAAGGAATTAGACACTGCTCAAAAGCACATTGACGCTGGTGCCAAGAAGGTTGTTAT CACTGCTCCATCTTCCACCGCCCCAATGTTCGTCATGGGTGTTAACGAAGAAAAATACACT TCTGACTTGAAGATTGTTTCCAACGCTTCTTGTACCACCAACTGTTTGGCTCCATTGGCC AAGGTTATCAACGATGCTTTCGGTATTGAAGAAGGTTTGATGACCACTGTCCACTCTTTG ACTGCTACTCAAAAGACTGTTGACGGTCCATCCCACAAGGACTGGAGAGGTGGTAGAA CCGCTTCCGGTAACATCATCCCATCCTCCACCGGTGCTGCTAAGGCTGTCGGTAAGG TCTTGCCAGAATTGCAAGGTAAGTTGAagGGTATGGCTTTCAGAGTCCCAACCGTCGAT GTCTCCGTTGTTGACTTGACTGTCAAGTTGAACAAGGAAACCACCTACGATGAAATCAA GAAGGTTGTTAAGGCTGCCGCTGAAGGTAAGTTGAAGGGTGTTTTGGGTTACACCGAA GACGCTGTTGTCTCCTCTGACTTCTTGGGTGACTCTCACTCTTCCATCTTCGATGCTTCC GCTGGTATCCAATTGTCTCCAAAGTTCGTCAAGTTGGTCTCCTGGTACGACAACGAATACG GTTACTCTACCAGAGTTGTCGACTTGGTTGAACACGTTGCCAAGGCTTAA |

Appendix 1 Continued on next page

*Appendix 1 Continued*

| Reagent type (species) or resource | Designation | Source or reference | Identifiers | Additional information |
|---|---|---|---|---|
| Sequence-based reagent | GPM1–CCmut | PMID:1386023 | | ATGCCAAAGTTAGTTTTAGTTAGACACGGTCAATCCGAATGGAACGAAAAGAACTTAT TCACCGGTTGGGTTGATGTTAAATTGTCTGCCAAGGGTCAACAAGAAGCCGCTAGAGC CGGTGAATTGTTGAAGGAAAAGAAGGTCTACCCAGACGTCTTGTACACTTCCAAGTT GTCCAGAGCTATCCAAACTGCTAACATTGCTTTGGAAAAGGCTGACAGATTATGGATTCCA GTCAACAGATCCTGGAGATTGAACGAAAGACATTACGGTGACTTACAAGGTAAGGACA AGGCTGAAACTTTGAAGAAGTTCGGTGAAGAAAAATTCAACACCTACAGAAGATCCTTC GATGTTCCACCTCCCCCAATCGACGCTTCTTCTCCATTCTCTCAAAAGGGTGATGAAAGA TACAAGTACGTTGACCCAAATGTCTTGCCAGAAACTGAATCTTTGGCTTTGGTCATTGACA GATTGTTGCCATACTGGCAAGATGTCATTGCCAAGGACTTGTTGAGTGGTAAGACCGT CATGATCGCCGCTgcCGGTAACTCCTTGAGAGGTTTGGTTAAGCACTTGGAAGGTATCTCT GATGCTGACATTGCTAAGTTGAACATCCCAACTGGTATTCCATTGGTCTTCGAATTGGACG AAAACTTGAAGCCATCTAAGCCATCTTACTACTTGGACCCAGAAGCTGCCGCTGCTGGT GCCGCTGCTGTTGCCAACCAAGGTAAGAAATAA |
| Sequence-based reagent | TDH3-CCmut | PMID:25580853 | | ATGGTTAGAGTTGCTATTAACGGTTTCGGTAGAATCGGTAGATTGGTCATGAGAATTGCT TTGTCTAGACCAAACGTCGAAGTTGTTGCTTTGAACGACCCATTCATCACCAACGACTACG CTGCTTACATGTTCAAGTACGACTCCACTCACGGTAGATACGCTGGTGAAGTTTCCCA CGATGACAAGCACATCATTGTCGATGGTAAGAAGATTGCTACTTACCAAGAAAGAGAC CCAGCTAACTTGCCATGGGGTTCTTCCAACGTTGACATCGCCATTGACTCCACTGGTG TTTTCAAGGAATTAGACACTGCTCAAAAGCACATTGACGCTGGTGCCAAGAAGGTTGTT ATCACTGCTCCATCTTCCACCGCCCCAATGTTCGTCATGGGTGTTAACGAAGAAAAATAC ACTTCTGACTTGAAGATTGTTTCCAACGCTTCTTcTACCACCAACTGTTTGGCTCCATTGGC CAAGGTTATCAACGATGCTTTCGGTATTGAAGAAGGTTTGATGACCACTGTCCACTCT TTGACTGCTACTCAAAAGACTGTTGACGGTCCATCCCACAAGGACTGGAGAGGTGGTA GAACCGCTTCCGGTAACATCATCCCATCCTCCACCGGTGCTGCTAAGGCTGTCGGTA AGGTCTTGCCAGAATTGCAAGGTAAGTTGACCGGTATGGCTTTCAGAGTCCCAACCG TCGATGTCTCCGTTGTTGACTTGACTGTCAAGTTGAACAAGGAAACCACCTACGATGAA ATCAAGAAGGTTGTTAAGGCTGCCGCTGAAGGTAAGTTGAAGGGTGTTTTGGGTTAC ACCGAAGACGCTGTTGTCTCCTCTGACTTCTTGGGTGACTCTCACTCTTCCATCTTCGA TGCTTCCGCTGGTATCCAATTGTCTCCAAAGTTCGTCAAGTTGGTCTCCTGGTACGACAA CGAATACGGTTACTCTACCAGAGTTGTCGACTTGGTTGAACACGTTGCCAAGGCTTAA |
| Commercial assay, kit | Qubit RNA BR assay kit | ThermoFisher Scientific | Q10210 | RNA Quantification |
| Commercial assay, kit | EzLabel FluoroNeo | ATTO | WSE-7010 | Protein labeling |
| Commercial assay, kit | NuPAGE LDS sample buffer | ThermoFisher Scientific | NP0007 | Protein extraction |
| Commercial assay, kit | NuPAGE Bis-Tris Mini Protein Gels, 4–12%, 1.0 mm | ThermoFisher Scientific | NP0322BOX | |
| Commercial assay, kit | NuPAGE MOPS SDS Running Buffer(20 X) | ThermoFisher Scientific | NP0001 | |
| Commercial assay, kit | Qubit RNA BR assay kit | ThermoFisher Scientific | Q10210 | Quantification of RNA quantity |
| Commercial assay, kit | Extracellular OCR Plate Assay Kit | DOJINDO | E297 | Mesurement of oxygen consumption |
| Commercial assay, kit | Immobilon -P membrane, PVDF, 0.45 µm, 26 x 26 cm sheet | Millipore | #IPVH304F0 | |
| Chemical compound, drug | Hydrogen peroxide 30% | Santoku chemical industries | | |
| Chemical compound, drug | D(-)-sorbitol | Wako | | |
| Chemical compound, drug | Aureobasidin A | TAKARA | | |
| Chemical compound, drug | Rapamycin | Wako | | |
| Chemical compound, drug | Glycerol | nacalai tesque | | |
| Chemical compound, drug | Antimycin A | Abcam | | |
| Chemical compound, drug | Chemiluminescent substrates | Millipore | #WBLUF0100 | |
| Antibody | α-Atg13 | PMID:10995454 | | 1:3000 |
| Antibody | peroxidase-conjugated goat anti-rabbit secondary antibodies | Jackson ImmunoResearch | #111–035–003 | 1:10000 |
| Software, algorithm | FastP (0.20.0) | PMID:30423086 | | |
| Software, algorithm | Hisat2 (2.2.0) | PMID:31375807 | | |
| Software, algorithm | Samtools (1.11) | PMID:19505943 | | |
| Software, algorithm | Stringtie (2.1.2) | PMID:25690850 | | |
| Software, algorithm | EdgeR (3.28.1) | PMID:19910308 | | |
| Software, algorithm | YeastSpotter | PMID:31095270 | | |

*Appendix 1 Continued on next page*

*Appendix 1 Continued*

| Reagent type (species) or resource | Designation | Source or reference | Identifiers | Additional information |
|---|---|---|---|---|
| Software, algorithm | Cellprofiler (4.2.6) | PMID:17269487 | | |
| Software, algorithm | Image Quant TL | GE Healthcare | | version 4.0.7 |
| Software, algorithm | LASX | Leica | | |
| Software, algorithm | Proteomaps | PMID:24889604 | | http://bionic-vis.biologie.uni-greifswald.de/ |
| Other | LAS-4000 | GE Healthcare | | |
| Other | DMI6000B | Leica | | |
| Other | MultiNA | Shimazu | #TVS062CA | |
| Other | COMPACT ROCKING INCUBATOR | ADVANTEC | | |
| Other | Infinite F200PRO | TECAN | | |
| Other | MegaLight 100 | SCHOTT | | |
| Other | Mini Gel Tank | ThermoFisher Scientific | #A25977 | |
| Other | Light–Capture II | ATTO | | |
| Sequence-based reagent (primer) | Y66G_f | this paper | | CCTTAGTCACTACTTTAACTggTGGTGTTCAATcTTTTTCTA |
| Sequence-based reagent (primer) | Y66G_r | this paper | | TAGAAAAAgATTGAACACCAccAGTTAAAGTAGTGACTAAGG |
| Sequence-based reagent (primer) | mCherry_Y72G_f | this paper | | TTTTGTCACCACAATTTATGggTGGTTCAAAAGCTTATGTTA |
| Sequence-based reagent (primer) | mCherry_Y72G_r | this paper | | TAACATAAGCTTTTGAACCAccCATAAATTGTGGTGACAAAA |
| Sequence-based reagent (primer) | mCherry-Kafri_Y72G_f | this paper | | CTGTCCCCTCAGTTCATGggCGGCTCCAAGGCCTACGTGAAG |
| Sequence-based reagent (primer) | mCherry-Kafri_Y72G_r | this paper | | CTTCACGTAGGCCTTGGAGCCGccCATGAACTGAGGGGACAG |
| Sequence-based reagent (primer) | moxGFP+Y_f | this paper | | GTATGGATGAATTGTACAAATATTAAgtgaatttactttaaat |
| Sequence-based reagent (primer) | moxGFP+Y_r | this paper | | atttaaagtaaattcacTTAATATTTGTACAATTCATCCATAC |
| Sequence-based reagent (primer) | T65S_f | this paper | | TTAGTCACTACTTTATcTTATGGTGTTCAATcT |
| Sequence-based reagent (primer) | T65S_r | this paper | | AgATTGAACACCATAAgATAAAGTAGTGACTAA |
| Sequence-based reagent (primer) | ACT1pro_f | this paper | | TGATGAGAGCCAGCTTAAAGAGTTAAAAATTTCATAGCTAaacaccggtgggggctgctg |
| Sequence-based reagent (primer) | ACT1pro_r | this paper | | ACAGCTTCACCCTTAGAAACCATtgttaattcagtaaattttcgatct |
| Sequence-based reagent (primer) | TDH3pro_f | this paper | | TGATGAGAGCCAGCTTAAAGAGTTAAAAATTTCATAGCTAgtaagggagttagaatcattttg |
| Sequence-based reagent (primer) | TDH3pro_r | this paper | | GTATGGACGAATTGTACAAGTAAgtgaatttactttaaatcttgcatt |
| Sequence-based reagent (primer) | YAP5pro_f | this paper | | TAACAGCTTCACCCTTAGAAACCATgactgtgataatatgctagttacac |
| Sequence-based reagent (primer) | YAP5pro_r | this paper | | TAACAGCTTCACCCTTAGAAACCATgactgtgataatatgctagttacac |
| Sequence-based reagent (primer) | PHO84pro_f | this paper | | TGATGAGAGCCAGCTTAAAGAGTTAAAAATTTCATAGCTAcacttcgttttttaccgtttagta |
| Sequence-based reagent (primer) | PHO84pro_r | this paper | | TAACAGCTTCACCCTTAGAAACCATttggattgtattcgtggagttttgt |
| Sequence-based reagent (primer) | ZPS1pro_f | this paper | | TGATGAGAGCCAGCTTAAAGAGTTAAAAATTTCATAGCTActtcttgctagtatatgacatac |

*Appendix 1 Continued on next page*

*Appendix 1 Continued*

| Reagent type (species) or resource | Designation | Source or reference | Identifiers | Additional information |
|---|---|---|---|---|
| Sequence-based reagent (primer) | ZPS1pro_r | this paper | | TAACAGCTTCACCCTTAGAAACCATaatgtttagtagttgtgtgtggatt |
| Sequence-based reagent (primer) | ADE17pro_f | this paper | | TGATGAGAGCCAGCTTAAAGAGTTAAAAATTTCATAGCTAtcgttatggaggtatagaaatgaa |
| Sequence-based reagent (primer) | ADE17pro_r | this paper | | TAACAGCTTCACCCTTAGAAACCATatttgatggtgatatgtgctttgat |
| Sequence-based reagent (primer) | CTR1pro_f | this paper | | TGATGAGAGCCAGCTTAAAGAGTTAAAAATTTCATAGCTAtttttccgcaaggccgcattttgaa |
| Sequence-based reagent (primer) | CTR1pro_r | this paper | | TAACAGCTTCACCCTTAGAAACCATtttgaatgtcaaatataatacactt |
| Sequence-based reagent (primer) | FIT2pro_f | this paper | | TGATGAGAGCCAGCTTAAAGAGTTAAAAATTTCATAGCTAtaccgaaatgacgaaatatactg |
| Sequence-based reagent (primer) | FIT2pro_r | this paper | | TAACAGCTTCACCCTTAGAAACCATtattattgttttgtgatggctttat |
| Sequence-based reagent (primer) | HXT7pro_f | this paper | | TGATGAGAGCCAGCTTAAAGAGTTAAAAATTTCATAGCTAaatagtactctcatcgctaagat |
| Sequence-based reagent (primer) | HXT7pro_r | this paper | | TAACAGCTTCACCCTTAGAAACCATtttttgattaaaattaaaaaaacttttttgtttttgtg |
| Sequence-based reagent (primer) | CUP1pro_f | this paper | | TGATGAGAGCCAGCTTAAAGAGTTAAAAATTTCATAGCTAgtcttttgctggcatttcttctaga |
| Sequence-based reagent (primer) | CUP1pro_r | this paper | | TAACAGCTTCACCCTTAGAAACCATgctgaatattttatgtgatgattga |
| Sequence-based reagent (primer) | OM14pro_f | this paper | | TGATGAGAGCCAGCTTAAAGAGTTAAAAATTTCATAGCTAtaactggtataattcgtttctcatg |
| Sequence-based reagent (primer) | OM14pro_r | this paper | | TAACAGCTTCACCCTTAGAAACCATattatgagatgctggaggtagatgt |
| Sequence-based reagent (primer) | OM45pro_f | this paper | | TGATGAGAGCCAGCTTAAAGAGTTAAAAATTTCATAGCTAtaaagataacaaattatcagacatg |
| Sequence-based reagent (primer) | OM45pro_r | this paper | | TAACAGCTTCACCCTTAGAAACCATccttatctgcttgtttttattaaatg |
| Sequence-based reagent (primer) | AGP1pro_f | this paper | | TGATGAGAGCCAGCTTAAAGAGTTAAAAATTTCATAGCTAtatcctagagcccaatgttccatga |
| Sequence-based reagent (primer) | AGP1pro_r | this paper | | TAACAGCTTCACCCTTAGAAACCATtgtgcgaagctatctttgtctatat |
| Sequence-based reagent (primer) | GAP1pro_f | this paper | | TGATGAGAGCCAGCTTAAAGAGTTAAAAATTTCATAGCTAcattgatagataaatcaacacagaa |
| Sequence-based reagent (primer) | GAP1pro_r | this paper | | TAACAGCTTCACCCTTAGAAACCATtttttatttctttttttttgtttcttataaatgttgctgtc |
| Sequence-based reagent (primer) | HXT1pro_f | this paper | | TGATGAGAGCCAGCTTAAAGAGTTAAAAATTTCATAGCTAgccacaatgaaacttcaattcatat |
| Sequence-based reagent (primer) | HXT1pro_r | this paper | | TAACAGCTTCACCCTTAGAAACCATgattttacgtatatcaactagttga |
| Sequence-based reagent (primer) | RIP1pro_f | this paper | | TGATGAGAGCCAGCTTAAAGAGTTAAAAATTTCATAGCTAgtcatgtatttctttccgctttagg |
| Sequence-based reagent (primer) | RIP1_r | this paper | | TAACAGCTTCACCCTTAGAAACCATaccaacaatgaccttatcaccatc |
| Sequence-based reagent (primer) | NSR1pro_f | this paper | | TGATGAGAGCCAGCTTAAAGAGTTAAAAATTTCATAGCTAttccaaactggttcattgaaatagg |
| Sequence-based reagent (primer) | NSR1_r | this paper | | TAACAGCTTCACCCTTAGAAACCATatcaaatgttttctttgaac |
| Sequence-based reagent (primer) | mScarlet-I_f | this paper | | ATGGTTTCTAAGGGTGAAGC |
| Sequence-based reagent (primer) | HIS3MX6_r | this paper | | TATATAAAATTAAATACGTAAATACAGCGTGCTGCGTGCTagctcgtttaaactggatgg |
| Sequence-based reagent (primer) | ATG13pro_f | this paper | | cggccgctctagaactagtGGATCCGATGCCTACGAAGATGATTC |
| Sequence-based reagent (primer) | ATG13ter_r | this paper | | attgggtaccgggccccccCTCGAGACGCAGTCAGCGGGTGACAA |

