## [Editor Report · eLife assessment]

This **convincing** study advances our understanding of the physiological consequences of the strong overexpression of non-toxic proteins in baker's yeast. The findings suggest that a massive protein burden results in nitrogen starvation and a shift in metabolism likely regulated via the TORC1 pathway, as well as defects in ribosome biogenesis in the nucleolus. The study presents findings and tools that are **important** for the cell biology and protein homeostasis fields.

---

## [Referee Report · Reviewer #1 (Public Review)]

Summary:

The study "Impact of Maximal Overexpression of a Non-toxic Protein on Yeast Cell Physiology" by Fujita et al. aims to elucidate the physiological impacts of overexpressing non-toxic proteins in yeast cells. By identifying model proteins with minimal cytotoxicity, the authors claim to provide insights into cellular stress responses and metabolic shifts induced by protein overexpression.

Strengths:

The study introduces a neutrality index to quantify cytotoxicity and investigates the effects of protein burden on yeast cell physiology. The study identifies mox-YG (a non-fluorescent fluorescent protein) and Gpm1-CCmut (an inactive glycolytic enzyme) as proteins with the lowest cytotoxicity, capable of being overexpressed to more than 40% of total cellular protein while maintaining yeast growth. Overexpression of mox-YG leads to a state resembling nitrogen starvation probably due to TORC1 inactivation, increased mitochondrial function, and decreased ribosomal abundance, indicating a metabolic shift towards more energy-efficient respiration and defects in nucleolar formation.

Weaknesses:

While the introduction of the neutrality index seems useful to differentiate between cytotoxicity and protein burden, the biological relevance of the effects of overexpression of the model proteins is unclear.

---

## [Referee Report · Reviewer #2 (Public Review)]

Summary:

In this manuscript, Fujita et al. characterized the neutrality indexes of several protein mutants in *S. cerevisiae* and uncovered that mox-YG and Gpm1-CCmut can be expressed as abundant as 40% of total proteins without causing severe growth defects. The authors then looked at the transcriptome and proteome of cells expressing excess mox-YG to investigate how protein burden affects yeast cells. Based on RNA-seq and mass-spectrometry results, the authors uncover that cells with excess mox-YG exhibit nitrogen starvation, respiration increase, inactivated TORC1 response, and decreased ribosomal abundance. The authors further showed that the decreased ribosomal amount is likely due to nucleoli defects, which can be partially rescued by nuclear exosome mutations.

Strengths:

Overall, this is a well-written manuscript that provides many valuable resources for the field, including the neutrality analysis on various fluorescent proteins and glycolytic enzymes, as well as the RNA-seq and proteomics results of cells overexpressing mox-YG. Their model on how mox-YG overexpression impairs the nucleolus and thus leads to ribosomal abundance decline will also raise many interesting questions for the field.

Weaknesses:

The authors concluded from their RNA-seq and proteomics results that cells with excess mox-YG expression showed increased respiration and TORC1 inactivation. I think it will be more convincing if the authors can show some characterization of mitochondrial respiration/membrane potential and the TOR responses to further verify their -omic results.

In addition, the authors only investigated how overexpression of mox-YG affects cells. It would be interesting to see whether overexpressing other non-toxic proteins causes similar effects, or if there are protein-specific effects. It would be good if the authors could at least discuss this point considering the workload of doing another RNA-seq or mass-spectrum analysis might be too heavy.

---

## [Referee Report · Reviewer #3 (Public Review)]

Summary:

Protein overexpression is widely used in experimental systems to study the function of the protein, assess its (beneficial or detrimental) effects in disease models, or challenge cellular systems involved in synthesis, folding, transport, or degradation of proteins in general. Especially at very high expression levels, protein-specific effects and general effects of a high protein load can be hard to distinguish. To overcome this issue, Fujita et al. use the previously established genetic tug-of-war system to identify proteins that can be expressed at extremely high levels in yeast cells with minimal protein-specific cytotoxicity (high 'neutrality'). They focus on two versions of the protein mox-GFP, the fluorescent version and a point mutation that is non-fluorescent (mox-YG) and is the most 'neutral' protein on their screen. They find that massive protein expression (up to 40% of the total proteome) results in a nitrogen starvation phenotype, likely inactivation of the TORC1 pathway, and defects in ribosome biogenesis in the nucleolus.

Strengths:

This work uses an elegant approach and succeeds in identifying proteins that can be expressed at surprisingly high levels with little cytotoxicity. Many of the changes they see have been observed before under protein burden conditions, but some are new and interesting. This work solidifies previous hypotheses about the general effects of protein overexpression and provides a set of interesting observations about the toxicity of fluorescent proteins (that is alleviated by mutations that render them non-fluorescent) and metabolic enzymes (that are less toxic when mutated into inactive versions).

Weaknesses:

The data are generally convincing, however in order to back up the major claim of this work - that the observed changes are due to general protein burden and not to the specific protein or condition - a broader analysis of different conditions would be highly beneficial.

Major points:

(1) The authors identify several proteins with high neutrality scores but only analyze the effects of mox/mox-YG overexpression in depth. Hence, it remains unclear which molecular phenotypes they observe are general effects of protein burden or more specific effects of these specific proteins. To address this point, a proteome (and/or transcriptome) of at least a Gpm1-CCmut expressing strain should be obtained and compared to the mox-YG proteome. Ideally, this analysis should be done simultaneously on all strains to achieve a good comparability of samples, e.g. using TMT multiplexing (for a proteome) or multiplexed sequencing (for a transcriptome). If feasible, the more strains that can be included in this comparison, the more powerful this analysis will be and can be prioritized over depth of sequencing/proteome coverage.

(2) The genetic tug-of-war system is elegant but comes at the cost of requiring specific media conditions (synthetic minimal media lacking uracil and leucine), which could be a potential confound, given that metabolic rewiring, and especially nitrogen starvation are among the observed phenotypes. I wonder if some of the changes might be specific to these conditions. The authors should corroborate their findings under different conditions. Ideally, this would be done using an orthogonal expression system that does not rely on auxotrophy (e.g. using antibiotic resistance instead) and can be used in rich, complex mediums like YPD. Minimally, using different conditions (media with excess or more limited nitrogen source, amino acids, different carbon source, etc.) would be useful to test the robustness of the findings towards changes in media composition.

(3) The authors suggest that the TORC1 pathway is involved in regulating some of the changes they observed. This is likely true, but it would be great if the hypothesis could be directly tested using an established TORC1 assay.

(4) The finding that the nucleolus appears to be virtually missing in mox-YG-expressing cells (Figure 6B) is surprising and interesting. The authors suggest possible mechanisms to explain this and partially rescue the phenotype by a reduction-of-function mutation in an exosome subunit. I wonder if this is specific to the mox-YG protein or a general protein burden effect, which the experiments suggested in point 1 should address. Additionally, could a mox-YG variant with a nuclear export signal be expressed that stays exclusively in the cytosol to rule out that mox-YG itself interferes with phase separation in the nucleus?

Minor points:

(5) It would be great if the authors could directly compare the changes they observed at the transcriptome and proteome levels. This can help distinguish between changes that are transcriptionally regulated versus more downstream processes (like protein degradation, as proposed for ribosome components).

---

## [Author Response]

The following is the authors’ response to the original reviews.

General response

(1) Evaluation of mitochondrial activity in mox-YG overexpression cells

To determine whether the observed “mitochondrial development” seen in transcriptomic, proteomic, and microscopic analyses corresponds to an actual phenotypic shift toward respiration, we measured oxygen consumption in mox-YG overexpression cells. The results showed that oxygen consumption rates were indeed elevated in these cells, suggesting a metabolic shift from fermentation toward respiration. These findings have been incorporated into the revised manuscript as new Figure 4E and Figure 4—figure supplement 9, along with the corresponding descriptions in the Results section.

(2) Evaluation of TORC1 Pathway Inactivation in mox-YG Overexpression Cells

While the proteomic response in mox-YG overexpression cells overlapped with known responses to TORC1 pathway inactivation, we had not obtained direct evidence that TORC1 activity was indeed reduced. To address this, we assessed TORC1 activity by testing the effect of rapamycin, a TORC1 inhibitor, and by attempting to detect the phosphorylation state of known TORC1 targets. Our results showed that mox-YG overexpressing cells exhibited reduced sensitivity to rapamycin compared to vector control cells, supporting the idea that TORC1 is already inactivated in the mox-YG overexpression condition.

In parallel, we attempted to detect phosphorylation of TORC1 targets Sch9 and Atg13 by Western blotting. Specifically, we tested several approaches: detecting phospho-Sch9 using a phospho-specific antibody, assessing the band shift of HA-tagged Sch9, and monitoring Atg13 band shift using an anti-Atg13 antibody. While we were unable to detect Sch9 phosphorylation, likely due to technical limitations, we finally succeeded in detecting Atg13 with the help of our new co-author, Dr. Kamada. However, we observed a marked reduction in Atg13 protein levels in mox-YG overexpression cells, making it difficult to interpret the biological significance of any apparent decrease in phosphorylation. Therefore, we decided not to pursue further experiments on TORC1 phosphorylation within the current revision period.

These findings have been summarized in new Figure 4—figure supplement 7, and the relevant description has been added to the Results section.

(3) Phenotypes of Gpm1-CCmut

We focused our initial analysis on the phenotypes of cells overexpressing mox-YG, the protein with the lowest Neutrality Index (NI) in our dataset, as a model of protein burden. However, it remained unclear to what extent the phenotypes observed in mox-YG overexpression cells are generalizable to protein burden as a whole. We agree with the reviewers’ suggestion that it is important to examine whether similar phenotypes are also observed in cells overexpressing Gpm1-CCmut, which was newly identified in this study as having a similarly low NI. We therefore performed validation experiments using Gpm1-CCmut overexpression cells to assess whether they exhibit the characteristic phenotypes observed in mox-YG overexpression cells. These phenotypes included: transcriptional responses, mitochondrial development, metabolic shift toward respiration, and nucleolar shrinkage.

As a result, mitochondrial development and nucleolar shrinkage were also observed in Gpm1-CCmut overexpression cells, consistent with mox-YG. In contrast, the transcriptional response associated with amino acid starvation and the metabolic shift toward respiration were not observed. Furthermore, an abnormal rounding of cell morphology—absent in mox-YG overexpression cells—was uniquely observed in Gpm1-CCmut cells. These results suggest that the phenotypes observed under mox-YG overexpression may comprise both general effects of protein burden and effects specific to the mox-YG protein. Alternatively, it is possible that Gpm1-CCmut imposes a different kind of constraint or toxicity not shared with mox-YG. In any case, these findings highlight that the full range of phenotypes associated with protein burden cannot yet be clearly defined and underscore the need for future analyses using a variety of “non-toxic” proteins.

Given that these results form a coherent set, we have relocated original Figure 3—which previously presented the NI values of Gpm1 and Tdh3 in the original version—to new Figure 6, which now includes all related phenotypic analyses. Correspondingly, we have added new Figures 6—figure supplement 1 through 6—figure supplement 7. The associated results have been incorporated into the Results section, and we have expanded the Discussion to address this point

As a result of these revisions, the order of figures has changed from the original version. The correspondence between the original and revised versions is as follows:

original→ Revised

Figure 1 → Figure 1

Figure 2 → Figure 2

Figure 3 → Figure 6

Figure 4 → Figure 3

Figure 5 → Figure 4

Figure 6 → Figure 5

**Public Reviews:**

**Reviewer #1 (Public Review):**
Weaknesses:While the introduction of the neutrality index seems useful to differentiate between cytotoxicity and protein burden, the biological relevance of the effects of overexpression of the model proteins is unclear.

Thank you for your comment. This point is in fact the core message we wished to convey in this study. We believe that every protein possesses some degree of what can be described as “cytotoxicity,” and that this should be defined by the expression limit—specifically, the threshold level at which growth inhibition occurs. This index corresponds to what we term the neutrality index. We further argue that protein cytotoxicity arises from a variety of constraints inherent to each protein. These constraints act in a stepwise manner to determine the expression limit (i.e., the neutrality) of a given protein (Figure 1A). To demonstrate the real existence of such constraints, there are two complementary approaches: an inductive one that involves large-scale, systematic investigation of naturally occurring proteins, and a deductive one that tests hypotheses using selected model proteins. Our current study follows the latter approach. In addition, we define protein burden as a phenomenon that can only be elicited by proteins that are ultimately harmless (Figure 1B). We assume that such burden results in a shared physiological state, such as depletion of cellular resources. Through continued efforts to identify a protein suitable for investigating this phenomenon, we eventually arrived at mox-YG. As the reviewer rightly pointed out, examining only mox-YG does not reveal the full picture of protein burden. In fact, in response to the reviewer’s suggestion, we investigated the physiological consequences of overexpressing a mutant glycolytic protein, Gpm1-CCmut (General Response 3). We found that the resulting phenotype was notably different from that observed in cells overexpressing mox-YG. Going forward, we believe that our study provides a foundation for further systematic exploration of “harmless proteins” and the cellular impacts of their overexpression.

**Reviewer #2 (Public Review):**
Weaknesses:The authors concluded from their RNA-seq and proteomics results that cells with excess mox-YG expression showed increased respiration and TORC1 inactivation. I think it will be more convincing if the authors can show some characterization of mitochondrial respiration/membrane potential and the TOR responses to further verify their -omic results.

These points are addressed in General Response 1 and 2.

In addition, the authors only investigated how overexpression of mox-YG affects cells. It would be interesting to see whether overexpressing other non-toxic proteins causes similar effects, or if there are protein-specific effects. It would be good if the authors could at least discuss this point considering the workload of doing another RNA-seq or mass-spectrum analysis might be too heavy.

These points are addressed in General Response 3.

**Reviewer #3 (Public Review):**
Weaknesses:The data are generally convincing, however in order to back up the major claim of this work - that the observed changes are due to general protein burden and not to the specific protein or condition - a broader analysis of different conditions would be highly beneficial.

These points are addressed in General Response 3.

Major points:(1) The authors identify several proteins with high neutrality scores but only analyze the effects of mox/mox-YG overexpression in depth. Hence, it remains unclear which molecular phenotypes they observe are general effects of protein burden or more specific effects of these specific proteins. To address this point, a proteome (and/or transcriptome) of at least a Gpm1-CCmut expressing strain should be obtained and compared to the mox-YG proteome. Ideally, this analysis should be done simultaneously on all strains to achieve a good comparability of samples, e.g. using TMT multiplexing (for a proteome) or multiplexed sequencing (for a transcriptome). If feasible, the more strains that can be included in this comparison, the more powerful this analysis will be and can be prioritized over depth of sequencing/proteome coverage.

This comment has been addressed in General Response 3. Gpm1-CCmut overexpression cells exhibited both phenotypes that were shared with, and distinct from, those observed in mox-YG overexpression cells. To define a unified set of phenotypes associated with "protein burden," we believe that extensive omics analyses targeting multiple "non-toxic" protein overexpression strains will be necessary. However, such an effort goes beyond the scope of the current study, and we would like to leave it as an important subject for future investigation.

(2) The genetic tug-of-war system is elegant but comes at the cost of requiring specific media conditions (synthetic minimal media lacking uracil and leucine), which could be a potential confound, given that metabolic rewiring, and especially nitrogen starvation are among the observed phenotypes. I wonder if some of the changes might be specific to these conditions. The authors should corroborate their findings under different conditions. Ideally, this would be done using an orthogonal expression system that does not rely on auxotrophy (e.g. using antibiotic resistance instead) and can be used in rich, complex mediums like YPD. Minimally, using different conditions (media with excess or more limited nitrogen source, amino acids, different carbon source, etc.) would be useful to test the robustness of the findings towards changes in media composition.

We appreciate the reviewer’s clear understanding of both the advantages and limitations of the gTOW system. As rightly pointed out, since our system relies on leucine depletion, it is essential to carefully consider the potential impact this may have on cellular metabolism. Another limitation—though it also serves as one of the strengths—of the gTOW system is its reliance on copy number variation to achieve protein overexpression. This feature limits the possibility of observing rapid responses, as immediate induction is not feasible. To address this issue, we have recently developed a strong and inducible promoter that minimizes effects on other metabolic systems (Higuchi et al., 2024), and we believe this tool will be essential in future experiments.

In response to the reviewer’s comments, we conducted two additional sets of experiments. First, we established a new overexpression system in nutrient-rich conditions (YPD medium) that is conceptually similar to gTOW but uses aureobasidin A and the *AUR1d* resistance gene to promote gene amplification (new Figure 4—figure supplement 2). Using this system, we observed that non-fluorescent YG mutants led to increased expression of mox. Total protein levels appeared to rise correspondingly, suggesting that the overall synthetic capacity of cells might be higher in YPD compared to SC medium. However, the degree of overexpression achieved in this system was insufficient to strongly inhibit growth, meaning we could not replicate the stress conditions observed with the original gTOW system. Further studies will be needed to determine whether stronger induction under these nutrient-rich conditions will yield comparable responses.

Second, we performed a control experiment to examine whether the amino acid starvation response observed in mox-YG overexpressing cells could be attributed to leucine depletion from the medium (new Figure 3—figure supplement 3). By titrating leucine concentrations in SC medium, we confirmed that lower leucine levels reduced the growth rate of vector control cells, indicating leucine limitation. However, *GAP1* induction was not observed under these conditions. In contrast, mox-YG overexpression led to strong *GAP1* induction under similar growth-inhibitory conditions, suggesting that the amino acid starvation response is not simply due to environmental leucine depletion, but rather a consequence of the cellular burden imposed by mox-YG overexpression.

These findings have been incorporated into the manuscript, along with the corresponding figures (new Figure 4—figure supplement 2, Figure 3—figure supplement 3), and relevant descriptions have been added to the Results and Discussion sections.

(3) The authors suggest that the TORC1 pathway is involved in regulating some of the changes they observed. This is likely true, but it would be great if the hypothesis could be directly tested using an established TORC1 assay.

This comment has been addressed in General Response 2. We assessed the rapamycin sensitivity of mox-YG overexpression cells—which was found to be reduced—and attempted to detect phosphorylation of the TORC1 target Atg13, although the latter was only partially successful. These findings have been incorporated into the Results section.

(4) The finding that the nucleolus appears to be virtually missing in mox-YG-expressing cells (Figure 6B) is surprising and interesting. The authors suggest possible mechanisms to explain this and partially rescue the phenotype by a reduction-of-function mutation in an exosome subunit. I wonder if this is specific to the mox-YG protein or a general protein burden effect, which the experiments suggested in point 1 should address. Additionally, could a mox-YG variant with a nuclear export signal be expressed that stays exclusively in the cytosol to rule out that mox-YG itself interferes with phase separation in the nucleus?

As also described in our General Response 3, we observed nucleolar shrinkage upon Gpm1-CCmut overexpression as well (new Figure 6E and 6—figure supplement 7), suggesting that this phenomenon may represent a general feature of protein burden. The reviewer’s suggestion to test whether this effect persists when mox-YG is excluded from the nucleus is indeed intriguing. However, based on our previous work, we have shown that overexpression of NES-tagged proteins (e.g., NES-EGFP) causes severe growth inhibition due to depletion of nuclear export factors (Kintaka et al., 2020). Unfortunately, this technical limitation makes it difficult for us to carry out the proposed experiment as suggested.

Minor points:(5) It would be great if the authors could directly compare the changes they observed at the transcriptome and proteome levels. This can help distinguish between changes that are transcriptionally regulated versus more downstream processes (like protein degradation, as proposed for ribosome components).

We also considered this point to be important, and therefore compared the transcriptomic and proteomic changes associated with mox-YG overexpression. However, somewhat unexpectedly, we found little correlation between these two layers of response. As shown in new Figure 3 and 4 (original Figures 4 and 5), while genes related to oxidative phosphorylation were consistently upregulated at both the mRNA and protein levels in mox-YG overexpressing cells, ribosomal proteins showed a discordant pattern: their mRNA levels were significantly increased, whereas their protein levels were significantly decreased.

Several factors may explain this discrepancy: (1) differences in analytical methods between transcriptomics and proteomics; (2) temporal mismatches arising from the dynamic changes in mRNA and protein expression during batch culture; and (3) the possibility that, under protein burden conditions, specific regulatory mechanisms may govern the selective translation or targeted degradation of certain proteins. However, at this point, we were unable to clearly determine which of these factors account for the observed differences.

For this reason, we did not originally include a global transcriptome–proteome comparison in the manuscript. In response to the reviewer’s comment, however, we have now included the comparison data (new Figure 4—figure supplement 3D).

**Recommendations for the authors:**

**Reviewer #1 (Recommendations for the authors):**
Major points:(1) While the study provides a detailed description of physiological changes, the underlying mechanisms remain speculative. For example, the exact reasons for nitrogen source depletion or increased respiration are unclear. The transcriptomic and proteomic data should be complemented by basic growth assay tests on rapamycin or glycerol to strengthen these observations.

This comment has been addressed in General Responses 1 and 2. We conducted oxygen consumption assays and growth assays in the presence of rapamycin, and incorporated these results into the revised version of the manuscript.

We also performed culture experiments using glycerol as a carbon source. However, both the vector control and mox-YG overexpression cells showed extremely poor growth. Although there was a slight difference between the two, we judged that it would be difficult to draw any meaningful conclusions from these results. Therefore, we have chosen not to include them in the main text (the data are attached below for reference).

**Author response image 1. sa4fig1:** 

(2) The study mainly focuses on two proteins, mox-YG/ FP proteins and Gpm1-CCmut. Did the authors look also at a broader range of proteins with varying degrees of cytotoxicity to validate the neutrality index and generalize their findings? Such as known cytotoxic proteins.

In our calculation of the Neutrality Index (NI), we use two parameters: the maximum growth rate (expressed as %MGR relative to the control) and the protein expression level. For the latter, we measure the abundance of the overexpressed protein as a percentage of total cellular protein, based on the assumption that the protein is expressed at a sufficiently high level to be detectable by SDS-PAGE. In our view, proteins typically regarded as “cytotoxic” cannot be overexpressed to levels detectable by SDS-PAGE without the use of more sensitive techniques such as Western blotting. This limitation in expression itself is an indication of their high cytotoxicity. Consequently, for such proteins, NI is determined solely by the MGR value, and will inherently fall below 100.

To test whether this interpretation is valid, we re-evaluated a group of EGFP variants previously reported by us to exhibit higher cytotoxicity than EGFP (Kintaka et al., 2016), due to overloading of specific cellular transport pathways. These include EGFPs tagged with localization signals. At the time of the original study, we had not calculated their NI values. Upon re-analysis, we found that all of these localization-tagged EGFP variants indeed have NI values below 100.

This result has been included as a new Figure 2—figure supplement 3, and the relevant descriptions have been added to the Results section.

(3) The partial rescue of ribosomal biosynthesis defects by a mutation in the nuclear exosome is intriguing but not fully explored. The specific role of the nuclear exosome in managing protein burden remains unclear. This result could be supported by alternative experiments. For example, would tom1 deletion or proteasome inhibition (degradation of ribosomal proteins in the nucleus) partially rescue the nuclear formation?

As described in the main text, our interest in exosome mutants was prompted by our previous SGA (Synthetic Genetic Array) analysis, in which these mutants exhibited positive genetic interactions with GFP overexpression—namely, they acted in a rescuing manner (Kintaka et al., 2020). In contrast, proteasome mutants did not show such positive interactions in the same screening. On the contrary, proteasome mutants that displayed negative genetic interactions have been identified, such as the *pre7ts* mutant. Furthermore, the proteasome is involved in various aspects of proteostasis beyond just orphan ribosomal proteins, making the interpretation of its effects potentially quite complex.

Regarding the *TOM1* mutant raised by the reviewer, we attempted to observe nucleolar morphology using the *NSR1-mScarlet-I* marker in the *tom1Δ* deletion strain. However, we were unsuccessful in constructing the strain. This failure may be due to the strong detrimental effects of this perturbation in the *tom1Δ* background. As we were unable to complete this experiment within the revision period, we would like to address this issue in future work.

Minor comments:(1) It would be interesting to include long-term cellular and evolutionary responses to protein overexpression to understand how cells adapt to chronic protein burden.

Thank you for the suggestion. We are currently conducting experiments related to these points. However, as they fall outside the scope of the present study, we would like to refrain from including the data in this manuscript.

(2) The microscopy of Nsr1 in Figure 6G does not clearly demonstrate the restored formation of the nucleolus in the mrt4-1 mutant. Electron microscopy images would be a better demonstration.

The restoration of nucleolar size in the *mtr4-1* mutant, as shown in Figure 5—figure supplement 5 (original Figure 6_S5), is statistically significant. However, as described in the main text, the degree of rescue by the mutation is partial, and, as the reviewer notes, not clearly distinguishable by eye. It becomes apparent only when analyzing a large number of cells, allowing for detection as a statistically significant difference. Given that electron microscopy images are inherently limited in the number of cells that can be analyzed and pose challenges for statistical evaluation, we believe it would be difficult to detect such a subtle difference using this method. Therefore, we respectfully ask for your understanding that we will not include additional EM experiments in this revision.

(3) On page 24, line 451 it says that of the 84 ribosomal proteins... latest reviews and structures described/ identified 79 ribosomal proteins in budding yeast of which the majority are incorporated into the pre-ribosomal particles in the nucleolus. We could not find this information in the provided reference. Please align with the literature.

Thank you for the comment. In *S. cerevisiae*, many ribosomal protein genes are duplicated due to gene duplication events, resulting in a total of 136 ribosomal proteins (http://ribosome.med.miyazaki-u.ac.jp/rpg.cgi?mode=genetable). However, not all of them are duplicated, and among the duplicated pairs, some can be distinguished by proteomic analysis based on differences in amino acid sequences, while others cannot. As a result, we report that 84 ribosomal proteins were “detected” in our proteomic analysis. To avoid confusion, we have added the following explanation to the legend of Figure 5—figure supplement 1 (original Figure 6_S1), as follows.

“Note that when the amino acid sequences of paralogs are identical, they cannot be distinguished by proteomic analysis, and the protein abundance of both members of the paralog pair is represented under the name of only one.”

**Reviewer #2 (Recommendations for the authors):**
(1) The authors mentioned that based on their proteomics results, overexpressing mox-YG appears to increase respiration. I think it is worth doing some quick verification, such as oxygen consumption experiments or mitochondrial membrane potential staining to provide some verification on that.

This comment has been addressed in General Response 1. We measured oxygen consumption in mox-YG overexpression cells and found that it was indeed elevated, suggesting a metabolic shift from fermentation toward aerobic respiration.

(2) Similar to point 1, the authors concluded from their proteomics data that the mox-YG overexpression induced responses that are similar to TORC1 inactivation. It might be worth testing whether there is any actual TORC1 inactivation, e.g. by detecting whether there is reduced Sch9 phosphorylation by western blot.

This comment has been addressed in General Response 2. We assessed the rapamycin sensitivity of mox-YG overexpression cells—which was found to be reduced—and attempted to detect phosphorylation of the TORC1 target Atg13, although the latter was only partially successful. These findings have been incorporated into the Results section.

(3) The authors showed that overexpressing excess mox-YG caused downregulated glycolysis pathways. It is worth discussing whether overexpressing glycolysis-related non-toxic proteins such as Gpm1-CCmut will also lead to similar results.

This comment has been addressed in General Response 3. Gpm1-CCmut overexpression cells exhibited both phenotypes shared with mox-YG overexpression and distinct ones. These findings suggest that a unified set of phenotypes associated with "protein burden" has yet to be clearly defined, and further investigation will be necessary to elucidate this.

**Reviewer #3 (Recommendations for the authors):**
(1) The authors identify several proteins with high neutrality scores but only analyze the effects of mox/mox-YG overexpression in depth. Hence, it remains unclear which molecular phenotypes they observe are general effects of protein burden or more specific effects of these specific proteins. To address this point, a proteome (and/or transcriptome) of at least a Gpm1-CCmut expressing strain should be obtained and compared to the mox-YG proteome. Ideally, this analysis should be done simultaneously on all strains to achieve a good comparability of samples, e.g. using TMT multiplexing (for a proteome) or multiplexed sequencing (for a transcriptome). If feasible, the more strains that can be included in this comparison, the more powerful this analysis will be and can be prioritized over depth of sequencing/proteome coverage.

This comment has been addressed in General Response 3. Gpm1-CCmut overexpression cells exhibited both phenotypes that were shared with, and distinct from, those observed in mox-YG overexpression cells. To define a unified set of phenotypes associated with "protein burden," we believe that extensive omics analyses targeting multiple "non-toxic" protein overexpression strains will be necessary. However, such an effort goes beyond the scope of the current study, and we would like to leave it as an important subject for future investigation.

(2) The genetic tug-of-war system is elegant but comes at the cost of requiring specific media conditions (synthetic minimal media lacking uracil and leucine), which could be a potential confound, given that metabolic rewiring, and especially nitrogen starvation are among the observed phenotypes. I wonder if some of the changes might be specific to these conditions. The authors should corroborate their findings under different conditions. Ideally, this would be done using an orthogonal expression system that does not rely on auxotrophy (e.g. using antibiotic resistance instead) and can be used in rich, complex mediums like YPD. Minimally, using different conditions (media with excess or more limited nitrogen source, amino acids, different carbon source, etc.) would be useful to test the robustness of the findings towards changes in media composition.

We appreciate the reviewer’s clear understanding of both the advantages and limitations of the gTOW system. As rightly pointed out, since our system relies on leucine depletion, it is essential to carefully consider the potential impact this may have on cellular metabolism. Another limitation—though it also serves as one of the strengths—of the gTOW system is its reliance on copy number variation to achieve protein overexpression. This feature limits the possibility of observing rapid responses, as immediate induction is not feasible. To address this issue, we have recently developed a strong and inducible promoter that minimizes effects on other metabolic systems (Higuchi et al., 2024), and we believe this tool will be essential in future experiments.

In response to the reviewer’s comments, we conducted two additional sets of experiments. First, we established a new overexpression system in nutrient-rich conditions (YPD medium) that is conceptually similar to gTOW but uses aureobasidin A and the *AUR1d* resistance gene to promote gene amplification (new Figure 4—figure supplement 2). Using this system, we observed that non-fluorescent YG mutants led to increased expression of mox. Total protein levels appeared to rise correspondingly, suggesting that the overall synthetic capacity of cells might be higher in YPD compared to SC medium. However, the degree of overexpression achieved in this system was insufficient to strongly inhibit growth, meaning we could not replicate the stress conditions observed with the original gTOW system. Further studies will be needed to determine whether stronger induction under these nutrient-rich conditions will yield comparable responses.

Second, we performed a control experiment to examine whether the amino acid starvation response observed in mox-YG overexpressing cells could be attributed to leucine depletion from the medium (new Figure 3—figure supplement 3). By titrating leucine concentrations in SC medium, we confirmed that lower leucine levels reduced the growth rate of vector control cells, indicating leucine limitation. However, *GAP1* induction was not observed under these conditions. In contrast, mox-YG overexpression led to strong *GAP1* induction under similar growth-inhibitory conditions, suggesting that the amino acid starvation response is not simply due to environmental leucine depletion, but rather a consequence of the cellular burden imposed by mox-YG overexpression.

These findings have been incorporated into the manuscript, along with the corresponding figures (new Figure 4—figure supplement 2, Figure 3—figure supplement 3), and relevant descriptions have been added to the Results and Discussion sections.

(3) The authors suggest that the TORC1 pathway is involved in regulating some of the changes they observed. This is likely true, but it would be great if the hypothesis could be directly tested using an established TORC1 assay.

This comment has been addressed in General Response 2. We assessed the rapamycin sensitivity of mox-YG overexpression cells—which was found to be reduced—and attempted to detect phosphorylation of the TORC1 target Atg13, although the latter was only partially successful. These findings have been incorporated into the Results section.

(4) The finding that the nucleolus appears to be virtually missing in mox-YG-expressing cells (Figure 6B) is surprising and interesting. The authors suggest possible mechanisms to explain this and partially rescue the phenotype by a reduction-of-function mutation in an exosome subunit. I wonder if this is specific to the mox-YG protein or a general protein burden effect, which the experiments suggested in point 1 should address. Additionally, could a mox-YG variant with a nuclear export signal be expressed that stays exclusively in the cytosol to rule out that mox-YG itself interferes with phase separation in the nucleus?

As also described in our General Response 3, we observed nucleolar shrinkage upon Gpm1-CCmut overexpression as well (new Figure 6E and 6—figure supplement 7), suggesting that this phenomenon may represent a general feature of protein burden. The reviewer’s suggestion to test whether this effect persists when mox-YG is excluded from the nucleus is indeed intriguing. However, based on our previous work, we have shown that overexpression of NES-tagged proteins (e.g., NES-EGFP) causes severe growth inhibition due to depletion of nuclear export factors (Kintaka et al., 2020). Unfortunately, this technical limitation makes it difficult for us to carry out the proposed experiment as suggested.

(5) It would be great if the authors could directly compare the changes they observed at the transcriptome and proteome levels. This can help distinguish between changes that are transcriptionally regulated versus more downstream processes (like protein degradation, as proposed for ribosome components).

We also considered this point to be important, and therefore compared the transcriptomic and proteomic changes associated with mox-YG overexpression. However, somewhat unexpectedly, we found little correlation between these two layers of response. As shown in new Figure 3 and 4 (original Figures 4 and 5), while genes related to oxidative phosphorylation were consistently upregulated at both the mRNA and protein levels in mox-YG overexpressing cells, ribosomal proteins showed a discordant pattern: their mRNA levels were significantly increased, whereas their protein levels were significantly decreased.

Several factors may explain this discrepancy: (1) differences in analytical methods between transcriptomics and proteomics; (2) temporal mismatches arising from the dynamic changes in mRNA and protein expression during batch culture; and (3) the possibility that, under protein burden conditions, specific regulatory mechanisms may govern the selective translation or targeted degradation of certain proteins. However, at this point, we were unable to clearly determine which of these factors account for the observed differences.

For this reason, we did not originally include a global transcriptome–proteome comparison in the manuscript. In response to the reviewer’s comment, however, we have now included the comparison data (new Figure 4—figure supplement 3D).

Minor points:(1) The authors repeatedly state that 'mitochondrial function' is increased. This is inaccurate in two ways: first, mitochondria have multiple functions, and it should be specified which one is referred to (probably mitochondrial respiration); second, the claim is based solely on the abundance of transcripts/proteins, which may or may not reflect increased activity.

The authors should either perform functional tests (e.g. measure oxygen consumption or extracellular acidification), or change their wording to more accurately reflect the findings.

To more directly reflect our findings, we revised two instances of the phrase “mitochondrial function” to “mitochondrial proteins” in the manuscript. Furthermore, as described in General Response 1, we confirmed that oxygen consumption is elevated in mox-YG overexpression cells. This observation suggests that mitochondrial respiratory activity is indeed enhanced under these conditions.

(2) Similarly, the authors state that FPs are 'not localized' (e.g. line 137). This should be specified (e.g. 'not actively sorted into cellular compartments other than the cytosol').

As pointed out by the reviewer, we have revised the relevant sections accordingly.

(3) In Figure 4D, some of the reporter assays don't fully recapitulate the RNAseq findings (e.g. for PHO84 and ZPS1, where mox-FS and mox-YG behave differently in the reporter assay, but not in the RNAseq data). This may stem from technical limitations given that the reporter assay relies on RFP expression which could generally be affected by protein overexpression (cf. ACT1pro in mox-FS), but it should be mentioned in the text.

We apologize for the confusion caused by our insufficient explanation of "moxFS" in new Figure 3D (original Figure 4D). As clarified here, "moxFS" refers to a frameshift mutant in which the mRNA is transcribed but the protein is not translated due to an early frameshift mutation. This is not a functional mox protein. The behavior of this mutant is nearly identical to that of the vector control, indicating that the transcriptional response observed in this assay is not triggered by mRNA expression itself, but rather by events occurring after protein synthesis begins. Importantly, the transcriptional responses identified by RNA-seq in mox-YG overexpression cells are largely recapitulated by this reporter assay, supporting the reliability of our experimental design.

We appreciate the reviewer’s comment, which helped us recognize the lack of clarity in our original description. In response, we have added an explanation of the FS mutation to the figure legend (new Figure 3D), and we have also expanded the description of the moxFS experimental results in the Results section.